# Tropical extreme droughts drive long-term increase in atmospheric $CO_2$ growth rate variability

Xiangzhong Luo [1,2,3]✉ & Trevor F. Keenan [1,2]✉

The terrestrial carbon sink slows the accumulation of carbon dioxide ($CO_2$) in the atmosphere by absorbing roughly 30% of anthropogenic $CO_2$ emissions, but varies greatly from year to year. The resulting variations in the atmospheric $CO_2$ growth rate (CGR) have been related to tropical temperature and water availability. The apparent sensitivity of CGR to tropical temperature ($\gamma_{CGR}^{T}$) has changed markedly over the past six decades, however, the drivers of the observation to date remains unidentified. Here, we use atmospheric observations, multiple global vegetation models and machine learning products to analyze the cause of the sensitivity change. We found that a threefold increase in $\gamma_{CGR}^{T}$ emerged due to the long-term changes in the magnitude of CGR variability (i.e., indicated by one standard deviation of CGR; $STD_{CGR}$), which increased 34.7% from 1960-1979 to 1985-2004 and subsequently decreased 14.4% in 1997-2016. We found a close relationship ($r^2 = 0.75$, $p < 0.01$) between $STD_{CGR}$ and the tropical vegetated area (23°S – 23°N) affected by extreme droughts, which influenced 6-9% of the tropical vegetated surface. A 1% increase in the tropical area affected by extreme droughts led to about 0.14 Pg C $yr^{-1}$ increase in $STD_{CGR}$. The historical changes in $STD_{CGR}$ were dominated by extreme drought-affected areas in tropical Africa and Asia, and semi-arid ecosystems. The outsized influence of extreme droughts over a small fraction of vegetated surface amplified the interannual variability in CGR and explained the observed long-term dynamics of $\gamma_{CGR}^{T}$.

[1] Climate and Ecosystem Sciences Division, Lawrence Berkeley National Laboratory, Berkeley, CA, USA. [2] Department of Environmental Science, Policy and Management, UC Berkeley, Berkeley, CA, USA. [3] Department of Geography, National University of Singapore, Singapore, Singapore. ✉email: xzluo.remi@nus.edu.sg; trevorkeenan@berkeley.edu

Year-to-year variations in the CGR ($\Delta$CGR; detrended CGR) mainly reflect changes in the terrestrial carbon sink[1], with a relatively small contribution from changes in ocean uptake and land use emissions[2–4] but see [5]. $\Delta$CGR is highly variable, ranging between $-2.0$ and $2.5$ PgC yr$^{-1}$ [6], and the majority of that variability is driven by changes in tropical climate[6–10] through the influence of climate on the tropical terrestrial ecosystems[2,11,12], among which tropical forests[1] or semi-arid ecosystems[13,14] (i.e., mostly the semi-arid ecosystems in the tropics[15]) or both[15] have been reported to play the primary role. In particular, $\Delta$CGR is sensitive to changes in temperature and water storage over the tropical land surface[1,6,8,9,16]. Positive anomalies of mean annual temperature over tropical land ($\Delta$MAT) have been associated with a higher CGR, potentially due to the suppression of tropical photosynthesis[17,18] and/or enhanced respiration[17,19,20], as have drier years[9,16]. Notably, the apparent temperature sensitivity of CGR ($\gamma_{CGR}^{T}$, see Methods) has been found to exhibit long-term dynamics, and has doubled between 1960 and the past decade[8,10]. In contrast to $\gamma_{CGR}^{T}$, the dynamics of the apparent sensitivity of CGR to water ($\gamma_{CGR}^{W}$) are less well understood, with conflicting reports suggesting either a non-evident global water sensitivity at the global scale[6,21] or a strong coupling between $\Delta$CGR and satellite-derived terrestrial water storage[9] and lagged precipitation[16]. Recent evidence demonstrated that $\gamma_{CGR}^{T}$ and $\gamma_{CGR}^{W}$ are related due to the land-atmosphere coupling by soil moisture[22], suggesting a potential change in $\gamma_{CGR}^{W}$ over time.

The underlying mechanisms for long-term changes in the climate sensitivity of CGR (i.e., $\gamma_{CGR}^{T}$ and $\gamma_{CGR}^{W}$) remain elusive, as process-based models demonstrate different climate sensitivities from observations[8,9]. Empirical evidence suggests that $\gamma_{CGR}^{T}$ is higher in years with greater tropical aridity[8], implying a role of tropical water availability in modulating $\gamma_{CGR}^{T}$. Water availability is known to influence land-atmosphere $CO_2$ exchange at seasonal and annual timescales through stomatal responses to atmospheric water stress or the downregulation of plant metabolism due to soil moisture deficits[23,24]. Extreme water deficits can further induce changes in land-atmosphere $CO_2$ exchange over longer time scales, through lagged responses and legacy effects of terrestrial ecosystems (i.e. mortality[25–27], fire[28], recovery[29,30] and deadwood decomposition[31]). This hierarchy of water-related processes can modulate CGR and manifest as the changes in $\gamma_{CGR}^{T}$, especially under extreme drought conditions. In fact, recent evidence has suggested that extreme droughts over small areas in Amazon and Australia have a disproportionally large contribution to the global carbon cycle[32,33], highlighting a potential role of tropical extreme droughts in modulating $\gamma_{CGR}^{T}$.

Therefore, we hypothesize that the long-term changes in $\gamma_{CGR}^{T}$ and $\gamma_{CGR}^{W}$ over the past six decades are related to changes in extreme droughts over tropical vegetated lands (23°S–23°N) and our objective is to test the hypothesis. However, estimates of apparent climate sensitivities vary between methods and climate data used[6,8,9] (Supplementary Fig. 1), and this uncertainty hinders attribution. We thus focus our examination on long-term CGR variability (i.e., indicated by one standard deviation of CGR within a time frame of decades; STD$_{CGR}$), which underlies the dynamics of derived climate sensitivities (see Methods). Specifically, we examined the long-term changes in $\gamma_{CGR}^{T}$ and $\gamma_{CGR}^{W}$ estimated from multiple mainstream methods[8,9], and related the best estimates of $\gamma_{CGR}^{T}$ and $\gamma_{CGR}^{W}$ to STD$_{CGR}$. We further examined the relationship between STD$_{CGR}$ and tropical droughts, using several key indicators of tropical water availability, an ensemble of dynamic global vegetation models[34], and the FLUXCOM machine learning products[21] based on observations from the global FLUXNET network (see Methods).

## Results

**Long-term changes in CGR and its climate sensitivities.** We used nine competing methods to derive $\gamma_{CGR}^{T}$ and $\gamma_{CGR}^{W}$ from $\Delta$CGR for every 20-year moving window between 1959 and 2016 (Table 1; Supplementary Fig. 1; see Methods). Predictors considered include anomalies of tropical mean annual temperature ($\Delta$MAT), mean annual precipitation ($\Delta$MAP), mean shortwave radiation($\Delta$RAD), 4-month lagged precipitation ($\Delta$MAP$_{lag}$), reconstructed satellite-derived terrestrial water storage ($\Delta$TWS) and interactions of temperature and water proxies, in univariate or multivariate linear regression models (Table 1; Supplementary Fig. 1; see Methods). Following model selection based on minimizing predictor collinearity, which can cause artificial temporal changes in the derived coefficients, we quantified $\gamma_{CGR}^{T}$ and $\gamma_{CGR}^{W}$ based on a multivariate linear regression of $\Delta$CGR on $\Delta$MAT, $\Delta$MAP and $\Delta$RAD (model M1, Table 1, see methods). $\gamma_{CGR}^{T}$ was significant ($p < 0.05$) in every 20-year window (Fig. 1a), increasing threefold between 1960 and 1999 (1960−1979: $1.83 \pm 0.45$ PgC yr$^{-1}$ K$^{-1}$ (mean ± s.d.); 1980−1999: $5.49 \pm 0.53$ PgC yr$^{-1}$ K$^{-1}$), consistent with previous reports[8,10], and decreasing by 33.6% in the most recent two decades (1997−2016: $3.64 \pm 0.53$ PgC yr$^{-1}$ K$^{-1}$) (Fig. 1a). In contrast, $\gamma_{CGR}^{W}$ was not significant ($p > 0.05$) in most 20-year windows and the sensitivity of CGR to tropical TWS ($\gamma_{CGR}^{TWS}$) derived from competing models (M3, M5, M8) was also not significant (Fig. 1; Supplementary Fig. 1).

We examined the ability of 15 Dynamic Global Vegetation Models (DGVMs)[35] and 3 machine learning products from the FLUXCOM project[21] to characterize long-term changes in the

**Table 1 The performance of nine competing models (M1-M9) to derive the temperature sensitivity ($\gamma_{CGR}^{T}$) and the water sensitivity of CGR ($\gamma_{CGR}^{W}$).**

| Models | Time range | Adj. R² | AIC | VIF | Reference |
|---|---|---|---|---|---|
| M1: $\Delta CGR = \gamma_{CGR}^{T}\Delta MAT + \gamma_{CGR}^{W}\Delta MAP + \gamma_{CGR}^{R}\Delta RAD$ | 1959–2016 | 0.47 ± 0.09 | 46.9 ± 3.6 | 1.25 ± 0.08 | 8 |
| M2: $\Delta CGR = \gamma_{CGR}^{T}\Delta MAT + \gamma_{CGR}^{W}\Delta MAP + \gamma_{CGR}^{R}\Delta RAD + \gamma_{CGR}^{i}(\Delta MAT \times \Delta MAP)$ | 1959–2016 | 0.46 ± 0.10 | 47.8 ± 2.9 | 1.46 ± 0.16 | |
| M3: $\Delta CGR = \gamma_{CGR}^{T}\Delta MAT + \gamma_{CGR}^{W}\Delta TWS + \gamma_{CGR}^{R}\Delta RAD$ | 1980–2016 | 0.56 ± 0.04 | 44.6 ± 2.5 | 1.50 ± 0.22 | 9 |
| M4: $\Delta CGR = \gamma_{CGR}^{T}\Delta MAT + \gamma_{CGR}^{W}\Delta MAP_{lag} + \gamma_{CGR}^{R}\Delta RAD$ | 1960–2016 | 0.50 ± 0.16 | 45.0 ± 5.8 | 2.49 ± 0.46 | 9 |
| M5: $\Delta CGR = \gamma_{CGR}^{T}\Delta MAT + \gamma_{CGR}^{W}\Delta TWS + \gamma_{CGR}^{R}\Delta RAD + \gamma_{CGR}^{i}(\Delta MAT \times \Delta TWS)$ | 1980–2016 | 0.59 ± 0.04 | 44.0 ± 3.0 | 1.57 ± 0.29 | |
| M6: $\Delta CGR = \gamma_{CGR}^{T}\Delta MAT + \gamma_{CGR}^{W}\Delta MAP_{lag} + \gamma_{CGR}^{R}\Delta RAD + \gamma_{CGR}^{i}(\Delta MAT \times \Delta MAP_{lag})$ | 1960–2016 | 0.50 ± 0.18 | 45.4 ± 7.0 | 2.64 ± 0.43 | |
| M7: $\Delta CGR = \gamma_{CGR}^{T}\Delta MAT$ | 1959–2016 | 0.46 ± 0.13 | 45.6 ± 3.1 | – | 8, 10 |
| M8: $\Delta CGR = \gamma_{CGR}^{W}\Delta TWS$ | 1959–2016 | 0.31 ± 0.06 | 51.6 ± 2.2 | – | 9 |
| M9: $\Delta CGR = \gamma_{CGR}^{W}\Delta MAP_{lag}$ | 1980–2016 | 0.42 ± 0.17 | 46.4 ± 5.4 | – | 16 |

Akaike information criterion (AIC) indicate the parsimony of model and variance inflation factor (VIF) indicate the collinearity of predictors. The model performance is evaluated for every 20-year window, therefore the statistical indicators (i.e., R², AIC, and VIF) are the mean of models from every window and uncertainty is one standard deviation. The interaction terms were normalized before used in the regression models.

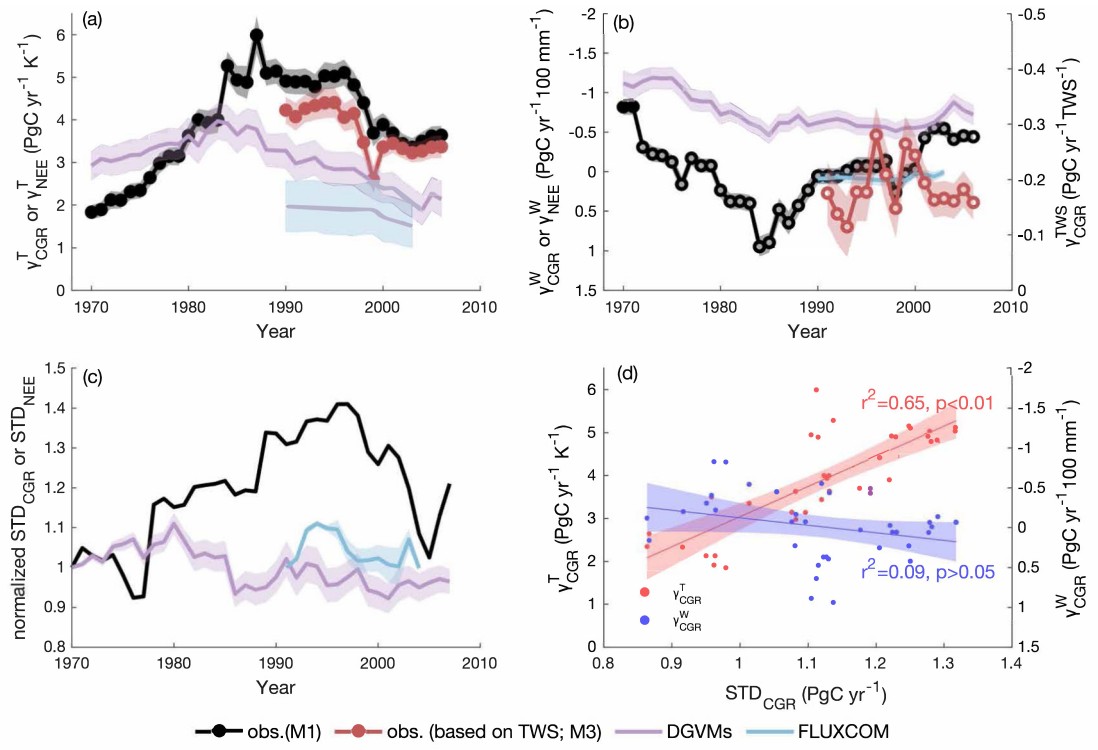

**Fig. 1 Temporal dynamics of the climate sensitivities of the atmospheric $CO_2$ growth rate (CGR) and net ecosystem exchange (NEE). a** Temporal dynamics of the apparent temperature sensitivity of observed CGR ($\gamma_{CGR}^T$) or modeled NEE ($\gamma_{NEE}^T$); **b** temporal dynamics of the apparent water sensitivity of CGR ($\gamma_{CGR}^W$) or modeled NEE ($\gamma_{NEE}^W$). The observed climate sensitivities are calculated using multivariate regressions of $\Delta$CGR to $\Delta$MAT and either $\Delta$MAP or $\Delta$TWS for each 20-year window from 1959 to 2016 (see Methods, model M1 and M3). A solid circle marker indicates significant ($p < 0.05$) sensitivities of CGR to climate variables in that 20-year window, while open circles indicate insignificant ($p > 0.05$) sensitivities. Black and red shaded areas indicate one standard error of climate sensitivities derived from 100 bootstrap estimates, considering the CGR uncertainty of 0.2 PgC $yr^{-1}$ [61]. Other shaded areas indicate the intermodel variations of climate sensitivity (i.e., one standard error). **c** The long-term dynamics of the variance of CGR ($STD_{CGR}$) and NEE ($STD_{NEE}$). $STD_{CGR}$ and $STD_{NEE}$ were calculated for every 20-year window from 1959 to 2016, and normalized by their respective first value (i.e., the $STD_{CGR}$ and $STD_{NEE}$ of 1959 to 1978). **d** The relationships between climate sensitivities of CGR (i.e., $\gamma_{CGR}^T$ and $\gamma_{CGR}^W$) and $STD_{CGR}$ (shading: 95% confidence interval). CGR in the years 1991–1993 are affected by the eruption of Mt Pinatubo and thus excluded.

CGR climate sensitivity. We found that both DGVMs and the machine learning products mischaracterized the temperature sensitivity ($\gamma_{NEE}^T$) of terrestrial net ecosystem exchange (NEE), the primary driver for the variation in CGR (Fig. 1a, b). $\gamma_{NEE}^T$ from DGVMs increased from $3.20 \pm 0.35$ PgC $yr^{-1}$ $K^{-1}$ in the 1970s to $3.65 \pm 0.42$ PgC $yr^{-1}$ $K^{-1}$ in the 1980s, and decreased to $2.19 \pm 0.37$ PgC $yr^{-1}$ $K^{-1}$ in the 2000s, however, the magnitude of change was much smaller than that of observed $\gamma_{CGR}^T$. Meanwhile, $\gamma_{NEE}^T$ from FLUXCOM was relatively constant, around $1.84 \pm 0.58$ PgC $yr^{-1}$ $K^{-1}$. Models such as those tested here are frequently used to infer the influence of soil moisture on global carbon cycle dynamics[13,14,21,23]. The discrepancy between observed $\gamma_{CGR}^T$ and modeled $\gamma_{NEE}^T$ we identified implies limitations in process-based models and machine learning methods, and calls into question their utility for diagnosing long-term changes in climate sensitivities.

Considering the DGVMs and FLUXCOM products we examined were forced by a similar climate dataset (i.e., CRU and CRU-NCEP, see Methods) that we used to obtain $\gamma_{CGR}^T$ and $\gamma_{CGR}^W$, the disagreement between the observed and modeled climate sensitivities can only be attributed to the difference between the observed variance of CGR (i.e. indicated by one standard deviation of CGR over the 20-year window; $STD_{CGR}$) and the modeled variance of NEE (i.e. indicated by one standard deviation of annual NEE over the 20-year window; $STD_{NEE}$). Indeed, we found that the $STD_{CGR}$ increased 34.7% from

0.98 PgC $yr^{-1}$ in 1960-1979 to 1.32 PgC $yr^{-1}$ in 1985–2004, and then decreased slightly by 14.4% to 1.13 PgC $yr^{-1}$ in 1997–2016 (Fig. 1c), and such a change in $STD_{CGR}$ underlies the dynamics of $\gamma_{CGR}^T$ we detected ($r^2 = 0.65$; Fig. 1d). The strong dependence of $\gamma_{CGR}^T$ on $STD_{CGR}$ was not affected by autocorrelations in the time series (Supplementary Fig. 3a, c). In contrast, $STD_{NEE}$ from DGVMs and FLUXCOM demonstrated no clear variation, with $STD_{NEE}$ fluctuating by only $-8$ to 10% over time (Fig. 1c).

**Tropical extreme droughts associated with changes in $STD_{CGR}$.** The long-term dynamics in $STD_{CGR}$ over the past 60 years were neither explained by the changes in the variability of ocean carbon uptake, emissions due to land use and land cover change[5,20] and emissions from fires[36] (Supplementary Fig. 2), nor by the estimates of NEE from DGVMs. Motivated by a previous study reporting the dependence of $\gamma_{CGR}^T$ on multiyear average aridity indexes for tropics[8], we examined the influence of long-term tropical water availability on $STD_{CGR}$. Our result showed that 20-year average tropical TWS, soil water content and mean annual precipitation ($\overline{TWS}$, $\overline{SWC}$, $\overline{MAP}$) were negatively and significantly related to the changes in $STD_{CGR}$ ($r^2 = 0.68$, 0.68, $0.65 \pm 0.09$, respectively; $p < 0.01$; Fig. 2a), showing that CGR was more variable in drier decades. In comparison, 20-year average MAT ($\overline{MAT}$) and vapor pressure deficit ($\overline{VPD}$) explained much less variance in $STD_{CGR}$ ($r^2 = 0.30 \pm 0.02$, 0.21, respectively;

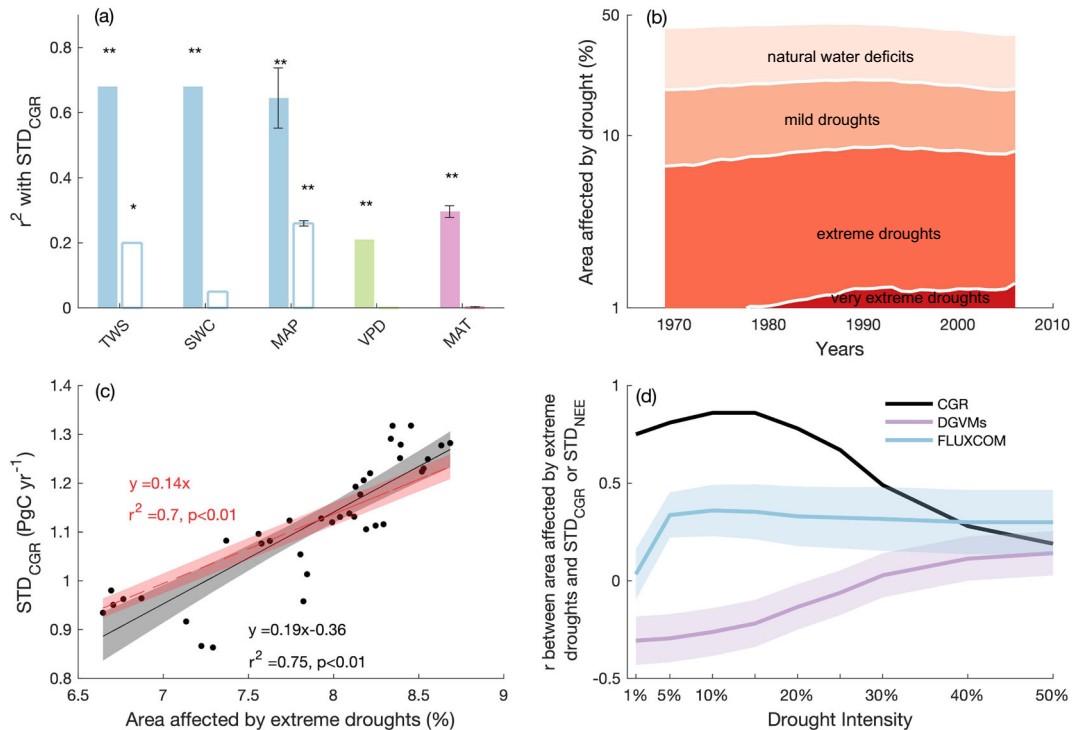

**Fig. 2 The relationships between the changes in STD$_{CGR}$ and tropical water availability and drought-affected area. a** The variance in STD$_{CGR}$ explained by long-term water availability or temperature in the tropics, as represented by 20-year average terrestrial water storage ($\overline{TWS}$), soil water content ($\overline{SWC}$), mean annual precipitation ($\overline{MAP}$), vapor pressure deficit ($\overline{VPD}$) and mean annual temperature ($\overline{MAT}$). The error bars indicate the uncertainty (one standard error) in $r^2$ when using alternative precipitation and temperature datasets (see Methods); the hollow bars indicate the variance in STD$_{CGR}$ explained by long-term tropical water availability or temperature, after accounting for autocorrelations in variables using the Cochrane-Ocrutt procedure (see Methods). *$p < 0.1$, **$p < 0.01$; note the correlations between STD$_{CGR}$ and $\overline{TWS}$, $\overline{SWC}$ and $\overline{MAP}$ are negative while the correlation between STD$_{CGR}$ and $\overline{VPD}$ and $\overline{MAT}$ are positive; **b** the temporal dynamics of the percentage of tropical vegetated area affected by droughts of different intensities; drought intensity is defined by the percentile of local monthly precipitation across the whole study period (1959–2016), where the bottom 1% precipitation indicates very extreme droughts, bottom 1–10% precipitation indicates extreme droughts, 10–25% precipitation indicates mild droughts, and 25–50% precipitation indicates natural water deficits; the y-axis is log-transformed; **c** The relationship between STD$_{CGR}$ and the tropical vegetated area affected by extreme and very extreme droughts. The shadings indicate the 95% confidence intervals of the linear regressions. The linear regressions with and without a y intercept were examined; **d** the correlation coefficient ($r$) between observed STD$_{CGR}$ or modeled (i.e., DGVMs and FLUXCOM) STD$_{NEE}$ and areas affected by droughts of different intensities in the tropics. Drought intensities are defined by the bottom percentiles (e.g., 1%, 10%, 25%) of monthly precipitation. Shaded areas indicate the intermodel variation of r (i.e., one standard error).

$p < 0.01$). We then removed the autocorrelations of the aforementioned 20-year average time series and found only the significant influence of $\overline{TWS}$ ($p < 0.1$) and $\overline{MAP}$ ($p < 0.01$) on STD$_{CGR}$ persisted (Fig. 2a; see Methods), highlighting a role of tropical water availability in modulating STD$_{CGR}$. Note that the removal of autocorrelation resulted in a smaller deterministic coefficient ($r^2$) between long-term water availability and STD$_{CGR}$, which is expected given the correction removes all the short-term variation in time series. As $\overline{MAP}$ showed the strongest explanatory power after considering autocorrelations in these explanatory time series, we used precipitation as the primary indicator for further analysis.

The observed negative couplings between tropical $\overline{TWS}$ and $\overline{MAP}$ and STD$_{CGR}$ (Fig. 2a) imply a considerable influence of tropical drought on long-term land-atmosphere $CO_2$ exchange. Following the definition of meteorological drought[37,38], we analysed drought occurrence in tropical regions over the past 60 years. We used percentiles of local monthly precipitation to classify different degrees of drought—from very extreme droughts (i.e., bottom 1% of precipitation), to extreme droughts (i.e., bottom 1–10%), to mild droughts (i.e., bottom 10–25%) and up to

natural water deficits (i.e., bottom 25–50%). We quantified the drought-affected area for each month, and calculated standardized annual drought-affected area using drought durations (see Methods). From 1959 to 2016, the tropical vegetated area affected by very extreme droughts increased from 0 to 1.4%, and the area affected by extreme droughts increased from 5.8% to 7.3% then slightly decreased to 6.7% (Fig. 2b). In contrast, the tropical area influenced by mild droughts decreased from 11.8% to 10.5% and the area influenced by natural water deficits decreased from 24.0% to 19.1% (Fig. 2b).

We examined the relationship between STD$_{CGR}$ and areas affected by different categories of drought, and found the changes in the extreme drought-affected area explained 75% of the variance in STD$_{CGR}$ ($p < 0.01$; Fig. 2c, d). Since the extreme droughts influenced around 6% to 9% of the tropical vegetated land surface, our result indicates a substantially outsized influence of extreme droughts on STD$_{CGR}$. To assess the robustness of the results, we removed the temporal autocorrelation in time series using two alternative methods and found that our conclusions are not qualitatively impacted by autocorrelation (Supplementary Fig. 3; see Methods). We found a 1% increase in

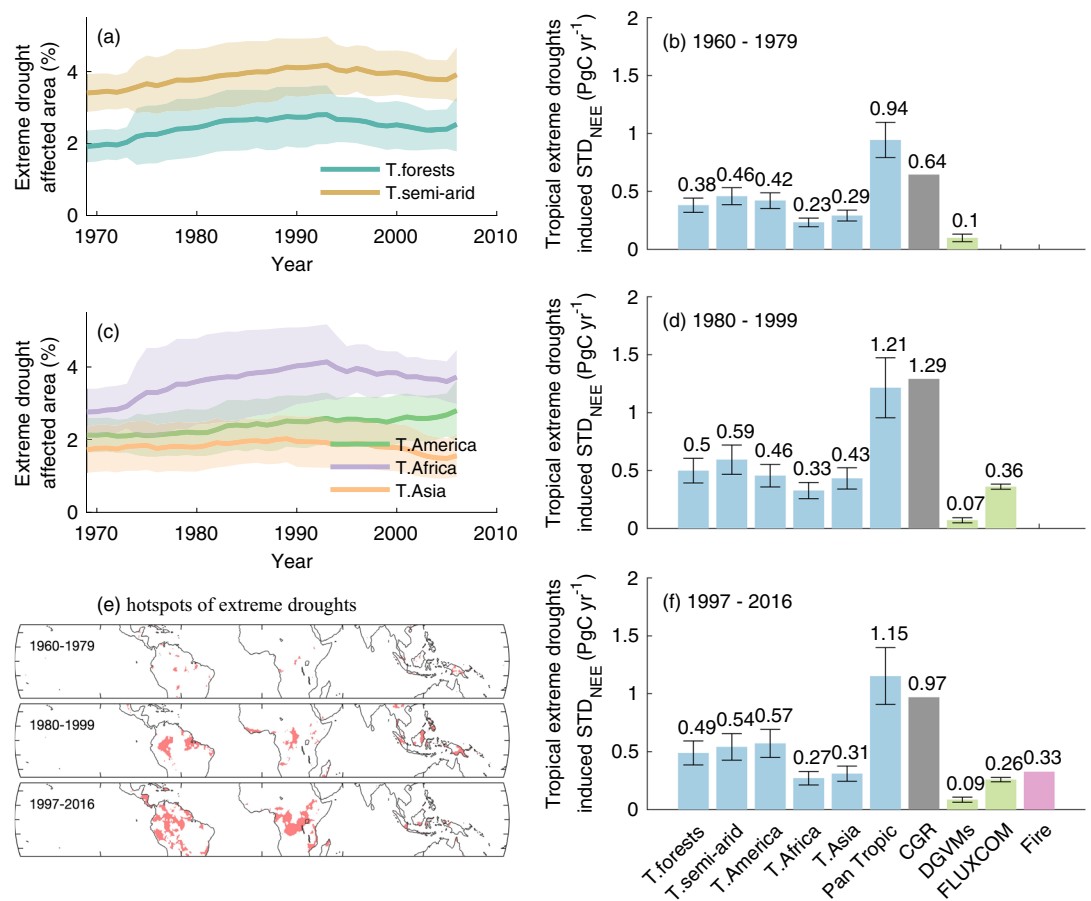

**Fig. 3 Spatial distribution of tropical extreme droughts and drought-induced changes in $STD_{CGR}$ over the past 60 years. a** the fraction of tropical vegetated area affected by extreme droughts for tropical (T.) forests and semi-arid regions. The solid line presents the mean fraction of tropical vegetated area affected by extreme droughts in every 20 years window from 1959 to 2016, with the x axis indicating the center of each window. The shaded area indicates the variability of the fraction (e.g., one standard deviation) in each 20-year window; **c** the fraction of tropical vegetated area affected by extreme droughts for tropical America, tropical Africa, and tropical Asia; **e** the changes of the extreme drought hotspots in three 20 years periods. Hotspots are defined as regions that are under extreme droughts for more than 10% of time; **b, d, f** The impacts of tropical extreme droughts on global net ecosystem exchange variability ($STD_{NEE}$) in three independent 20-year periods. The extreme drought-induced $STD_{NEE}$ was estimated using the relationship between drought-affected area and $STD_{CGR}$ (shown in Fig. 2c) in combination with a spatial weight based on FLUXCOM NEE (see Methods). The average impact from each region is indicated by the numbers in the figure, while the error bars indicate the variability of extreme drought-induced $STD_{NEE}$ in each 20-year window propagated from the variability in the fraction of area affected by extreme droughts shown in (**a**) and (**c**). The impact of droughts estimated by DGVMs and FLUXCOM are included as reference, where the error bars indicate the intermodel variations. Fire emissions are obtained from Global Fire Emissions Database (GFED4s).

extreme drought-affected area corresponded to a 0.14 Pg C $yr^{-1}$ increase in $STD_{CGR}$. In contrast, mild droughts had a limited influence on $STD_{CGR}$, as the addition of mild droughts reduced the coupling between drought-affected area and $STD_{CGR}$ (Fig. 2d). We repeated the analysis using $STD_{NEE}$ from DGVMs, and found models was unable to characterize the outsized influence of extreme droughts, with negative correlations between extreme drought-affected area and $STD_{NEE}$ across all drought categories (Fig. 2d; Supplementary Fig. 5b). In comparison, $STD_{NEE}$ estimated by FLUXCOM was positively but weakly related to the area affected by extreme droughts, implying that the data-driven NEE product was able to capture the influence of extreme droughts effect better than DGVMs, though with substantial underestimation of the effect (Fig. 2d; Supplementary Fig. 5c).

**Spatial variation in extreme drought-induced changes in $STD_{CGR}$.** Extreme droughts happened unevenly over time and space, and thus contributed differently to the variation of $STD_{CGR}$.

Among the three continents, tropical Africa had more area affected by extreme droughts (c. 3.62 ± 0.91% of tropical vegetated land surface) in the past 60 years than tropical America (c. 2.39 ± 0.63%) and tropical Asia (c. 1.82 ± 0.65%). The extreme drought-affected area in the tropical America showed a positive trend over time, while areas affected by droughts in tropical Africa and tropical Asia increased before the 1990s and then plateaued or decreased in the recent two decades (Fig. 3c), contributing to the recent decrease in $STD_{CGR}$. The areas most threatened by extreme droughts were located in tropical America and Africa, however, the location of drought hotspots demonstrated also long-term variations, as drought-affected areas were more concentrated in the 1980s and the 1990s than the recent two decades (Fig. 3e). In parallel, extreme droughts influenced more semi-arid ecosystems (c. 3.85 ± 0.72% of vegetated land surface) than forests (c. 2.47 ± 0.71%), and the drought-affected area of both ecosystems increased from the 1960s to the 1990s and decreased in the 2000s (Fig. 3a).

To further quantify the regional contributions of extreme droughts to $STD_{CGR}$, we applied the tight global relationship

between extreme drought-affected area and $STD_{CGR}$ (Fig. 2c) and a spatial weight based on average FLUXCOM NEE to partition the changes in $STD_{CGR}$ into the regional $STD_{NEE}$ of each continent (i.e., tropical America, Africa, and Asia) and biome (i.e., tropical semi-arid and tropical forests), assuming the temporal relationship holds in the spatial dimension (see Methods). Over the three independent 20-year periods, the extreme droughts in tropical semi-arid ecosystem contributed more to $STD_{NEE}$ than droughts in tropical forests. Among the three continents, droughts in tropical America incurred larger $STD_{NEE}$ than those in tropical Africa and Asia (Fig. 3b, d, f), however, the dynamics of overall $STD_{NEE}$ change were controlled by tropical Africa and Asia. The increase of $STD_{NEE}$ from 1960 to 1979 to 1980 to 1999 was about 0.27 Pg C $yr^{-1}$, and tropical Asia and Africa contributed 0.14 Pg C $yr^{-1}$ and 0.10 Pg C $yr^{-1}$, respectively. From 1980-1999 to 1997–2016, the decrease of $STD_{NEE}$ was about 0.06 Pg C $yr^{-1}$, which was a net balance of decreasing extreme droughts and $STD_{NEE}$ in tropical Africa and Asia—a total deduction of 0.17 Pg C $yr^{-1}$ and increasing extreme droughts and $STD_{NEE}$ in tropical America—a total increase of 0.11 Pg C $yr^{-1}$ in $STD_{NEE}$. In comparison, DGVMs showed limited influence of extreme droughts on $STD_{NEE}$ compared to $STD_{CGR}$ but FLUXCOM detected a decrease in $STD_{NEE}$ from 1980 to 1999 to 1997 to 2016.

## Discussion

In this study, we find there were long-term changes in CGR variability (i.e., $STD_{CGR}$), which increased then decreased in the past six decades. The changes in $STD_{CGR}$ were positively associated with changes in tropical extreme droughts and underlay previously unexplained dynamics of $\gamma_{CGR}^{T}$ [8,10]. We find that extreme drought-affected area, which accounted for only 6–9% of the tropical vegetated surface, explained 75% of the variance in $STD_{CGR}$. The historical increase and the recent decrease in $STD_{CGR}$ were dominated by drought area changes in tropical Africa and Asia, while tropical America showed monotonically increased drought area. In terms of biome contribution, tropical semi-arid ecosystems and forests contributed equally to the increase of $STD_{CGR}$ in the 1980s and the 1990s but semi-arid ecosystems dominated the decrease in $STD_{CGR}$ in the recent two decades. This study highlights the outsized role of tropical extreme droughts in influencing the long-term dynamics of CGR and amplifying $\gamma_{CGR}^{T}$, and has manifold implications for our current understanding of climate-carbon interactions.

Extreme events are known to influence the terrestrial carbon cycle[39], and drought is the most critical one[40]. Several studies have suggested that a few extreme events explained a significant amount of the variance in land-atmosphere carbon exchange, at seasonal or interannual time scales[41,42]. Here, our results show that extreme droughts influence CGR at the bi-decadal scale by amplifying $STD_{CGR}$ and $\gamma_{CGR}^{T}$. Extreme droughts induce various concurrent and lagged effects on ecosystems[43], which include processes of either carbon emissions or carbon uptake. Though we find a tight correlation between extreme drought-affected area and $STD_{CGR}$ (Fig. 2c), we acknowledge that the affected area is an integrated indicator of drought effects, which does not distinguish the various induced processes and their respective functioning time scales. Among the drought-induced effects, fire is unlikely to be the sole reason for the changes in $STD_{CGR}$ as our analysis does not show a change in the long-term variability in fire emissions (Supplementary Fig. 2), meaning the role of other process needs further investigations. We suggest that extreme droughts in semi-arid ecosystems and tropical Africa and Asia deserve more attention for understanding long-term dynamics of the terrestrial carbon cycle.

In this study, we use nine competing linear models to derive $\gamma_{CGR}^{T}$ and $\gamma_{CGR}^{W}$ (Supplementary Fig. 1). The values of $\gamma_{CGR}^{T}$ and

$\gamma_{CGR}^{W}$ and their long-term dynamics are highly influenced by the types of linear models and climate data used. Statistically, our result shows that $\gamma_{CGR}^{W}$ is insignificant as long as the models include tropical temperature as a predictor. However, when using univariate linear models we note $\gamma_{CGR}^{W}$ is significant (Supplementary Fig. 1h), caused by the strong correlation between $\Delta MAT$ and $\Delta TWS$[15] or tropical lagged precipitation[44]. Previous studies have reported nonlinear responses of the tropical terrestrial carbon fluxes to temperature[45], VPD[46], precipitation[47], and soil moisture[23], which question the common practices of using linear models to derive $\gamma_{CGR}^{T}$ and $\gamma_{CGR}^{W}$ [6,8,9], though nonlinear models may not be statistically stronger than linear models with less degrees of freedom for fitting. To avoid the uncertainties in climate sensitivities incurred by the type of models and data used, we use $STD_{CGR}$, which is calculated from perhaps the least uncertain term in the global carbon cycle (i.e., CGR), as a proxy for $\gamma_{CGR}^{T}$ to examine the long-term dynamics.

Tropical extreme droughts developed preferentially during El Niño events[48,49]. Therefore, the drought-$STD_{CGR}$ correlation can also be interpreted as an El Niño Southern Oscillation (ENSO)-$STD_{CGR}$ correlation. We use the Multivariate ENSO Index (MEI) to represent the frequency and strength of El Niño, and find MEI is positively related with $\Delta CGR$, $STD_{CGR}$ and the extreme drought-affected area ($p < 0.01$; Supplementary Fig. 4). The test shows ENSO not only modulates CGR at the interannual time scale[7], but also enhances the magnitude of CGR variability in periods with more frequent and stronger El Niño event by increasing extreme drought frequency. It is worth noting that MEI ($r^2 = 0.68$, Supplementary Fig. 4b) does not explain as much long-term variability in CGR as the extreme drought area ($r^2 = 0.75$, Fig. 2c).

DGVMs are unable to reproduce the tight positive relationship between $STD_{CGR}$ and tropical extreme drought-affected area (Fig. 2d; Supplementary Fig. 5), indicating questionable representation of extreme drought in models. Most DGVMs show a negative correlation between extreme drought-affected area and $STD_{NEE}$ (Fig. 2d, Supplementary Fig. 5b), which means extreme droughts cause limited land-atmosphere $CO_2$ exchange. This explains previously reported spatial asynchrony between carbon extremes and climate extremes in DGVMs[50], as carbon extremes in DGVMs are more likely driven by favorable climate than unfavorable climate extremes[51]. Missing components of drought–vegetation feedbacks (e.g., lagged effects) are potentially responsible for the biased estimates of NEE in DGVMs under extreme droughts. Previous studies have shown that in DGVMs droughts induce the largest impact during the climate extreme with little lagged influence[52], which is substantially shorter than observed drought effects lasting for several years[25,53]. With a focus only on transient stomatal control and soil moisture downregulattion[23], models generally lack enough mechanistic consideration of drought-induced legacy effects[27]. Moreover, ecosystems acclimate to temperature[54], $CO_2$[55], and extended droughts[56] to cope with water stress and adjust carbon uptake. Considering the increase in $CO_2$, temperature and drought frequency under future scenarios[38], these long-term trends indicate further biases in modeled $STD_{NEE}$.

FLUXCOM – a data-driven machine learning NEE product—was also unable to fully capture the dynamics in $\gamma_{CGR}^{T}$ and $STD_{CGR}$ (Fig. 1a, b). The underrepresentation of extreme droughts in the relatively short eddy covariance measurements (i.e., limited sites have more than 10 years of records)[57] and the lack of sites in the tropics[58] could lead to structural and long-term changes being undetected, and consequently to the underestimation of tropical NEE variability and CGR climate sensitivities. Other than the impact of extreme droughts, we acknowledge that the muted interannual variability of

FLUXCOM product may be also caused by the incapability of machine learning algorithms to capture low-frequency variations at the interannual time scale and the use of average remote sensing forcing[58]. However, unlike DGVMs, FLUXCOM identified a weak yet positive relationship between extreme drought-affected area and $STD_{NEE}$ (Fig. 2d), showing that it has a better representation of extreme drought effects than DGVMs. It is worth noting that NEE of tropical extreme drought-affected regions from FLUXCOM shows a sensitivity of $STD_{NEE}$ to the drought that is close to the sensitivity of $STD_{CGR}$ to drought (Supplementary Fig. 5). To further improve the prediction of machine learning algorithms of CGR variability, one potential path is to use algorithms such as Long Short-Term Memory (LSTM) to incorporate the lagged effects of climate extremes into the simulation of terrestrial carbon fluxes[59].

Water is elemental to the function of terrestrial ecosystems, and is known to influence the terrestrial carbon cycle through processes at multiple temporal and spatial scales. Our analysis shows that extreme drought-affected area in the tropics is positively associated with the changes in $STD_{CGR}$, which explains the pronounced variations in $\gamma_{CGR}^{T}$, as CGR is more variable in decades when there are more extreme droughts. Our findings provide a quantitative basis to examine the drought-$STD_{CGR}$ relationship, and provide a new perspective to understand carbon-water interactions over long periods.

## Methods

**Atmospheric $CO_2$ growth rate (CGR).** We used CGR retrieved from the global average atmospheric $CO_2$ concentration reported by the US National Oceanic and Atmospheric Administration Earth System Research Laboratory (NOAA/ESRL)[60] and collated by the Global Carbon Project[61], for the period between 1959 and 2016. NOAA/ESRL has measured $CO_2$ for several decades at a globally distributed network of air sampling sites, including 4 baseline observatories (i.e., Barrow, Mauna Loa, Samoa and South Pole) and 8 tall towers, air samples collected by volunteers at more than 50 sites, and air samples collected regularly from small aircraft mostly in North America (https://www.esrl.noaa.gov/gmd/ccgg/about.html). For the period before 1980, the annual global $CO_2$ concentration is constructed on the measurements of two long-term baseline sites (Mauna Loa and South Pole) reported by Scripps Institute of Oceanography[62]. After 1980, the annual global $CO_2$ concentration is constructed by averaging half-hourly latitudinal $CO_2$ concentrations interpolated from the measurements of all NOAA/ESRL sites where samples are predominantly of well-mixed marine boundary layer[60]. CGR was then calculated by multiplying a factor of 2.124 GtC ppm$^{-1}$ to the changes in atmospheric $CO_2$ concentration (ppm yr$^{-1}$)[63].

**Gridded Climate data.** Global monthly gridded air temperature, precipitation and solar radiation data at 0.5° were provided by the Climate Research Unit (CRU) at University of East Anglia[64]. We used the CRU TS 4.01 climate dataset which ranges from 1959 to 2016. Monthly vapor pressure deficit was calculated as the difference between saturated vapor pressure calculated based on monthly air temperature and actual vapor pressure provided by CRU. Monthly soil water content at 0.5° was calculated from CRU climate data using the simple process-led algorithms for simulating habitats (SPLASH)[65]. SPLASH models soil water content as the residual of precipitation, runoff and evapotranspiration, with evapotranspiration calculated using a Priestly-Taylor scheme[66]. From the resulting monthly gridded climate dataset, we calculated mean annual tropical temperature (MAT), precipitation (MAP), solar radiation (RAD) and soil water content (SWC) for vegetated land. The correlation between the temporal dynamics of MAT and MAP is −0.15 (Pearson's $r$, $p = 0.27$), between MAT and SWC, −0.23 ($p = 0.09$), between MAP and SWC, 0.68 ($p < 0.01$). In addition, we used several other gridded climate datasets produced based on global observations available from 1959 to 2016 to examine the robustness of our results. These gridded climate datasets are Berkeley Earth Surface Temperature (BEST)[67], CRUTEM4 surface temperature[68], NASA Goddard Institute for Space Studies surface temperature (GISS)[69], global temperature and precipitation produced by University of Delaware (UDEL)[70], global precipitation produced by the Global Precipitation Climatology Centre (GPCC)[71] and NOAA's PRECipitation REConstruction over Land (PRECL)[72].

**Ancillary remote sensing and modeled datasets.** We used a reconstructed terrestrial water storage (TWS) derived from the GRACE satellite as one of observational proxies for tropical water availability. The reconstructed TWS ranges from 1981 to 2017 and has a global coverage at 0.5°[9]. We extracted TWS data over tropical vegetated land surface for our analysis. We used the Moderate Resolution Imaging Spectroradiometer (MODIS) MOD12 land cover product[73] to delineate

biome types in the tropics—forests (i.e., evergreen broadleaf forests) and semi-arid ecosystems (i.e., grasslands, shrublands and savanna type ecosystems). We used the land cover type that was most prevalent during the period 2000–2013 for our study area. Global fire carbon emissions were acquired from Global Fire Emission Database (GFED4s)[36]. Two estimates of global carbon emissions due to land use and land cover (LULC) were included in our analysis. One is an estimate based on two bookkeeping models used in the Global Carbon Project[61], the other one is an estimate based on a process-based model that considers the influences of climate variation and dynamic biomass density in the LULC emission[5].

**Estimates of net ecosystem exchange (NEE).** We used annual NEE estimated from two sources, including (1) 15 dynamic global vegetation models (DGVMs) from TRENDY v6[35] participating in the Global Carbon Project[61] and (2) FLUX-COM fluxes upscaled from eddy-covariance measurements using three machine learning methods[21].

DGVMs used in this study include CABLE, CLASS-CTEM, CLM4.5(BGC), DLEM, ISAM, JSBACH, JULES, LPJ-GUESS, LPJ, LPX-Bern, OCN, ORCHIDEE, ORCHIDEE-MICT, SDGVM, VISIT (Supplementary Table 1). These models were driven either by monthly CRU or 6-hourly CRU-NCEP gridded climate dataset and dynamic atmospheric $CO_2$ concentrations. Some models consider the effects of $CO_2$ fertilization, LULC, and nitrogen deposition on the carbon cycle, but carbon emissions due to LULC were considered separately thus not included in the NEE output. We used the DGVMs annual NEE estimates from 1959 to 2016.

FLUXCOM[21] estimates global NEE by upscaling measurements from 224 eddy-covariance flux tower sites using three different machine learning algorithms: Random forests (RF), Artificial Neural Networks (ANN) and Multivariate Adaptive Regression Splines (MARS). Each machine learning algorithm was trained on daily fluxes using 11 inputs including site-level meteorological data and mean seasonal cycle of MODIS observations. After obtaining the trained algorithm, gridded climatic variables from CRU-NCEP and the mean seasonal cycle of MODIS data were used to produce carbon flux estimates on 0.5°×0.5° grids and at monthly intervals. We then summed up the monthly gridded data and got the FLUXCOM annual NEE estimates from 1980 to 2013.

**Climate sensitivity of CGR.** We calculated the temperature sensitivity ($\gamma_{CGR}^{T}$) and the water sensitivity of CGR ($\gamma_{CGR}^{W}$) based on univariate or multivariate linear regressions of the anomalies of CGR ($\Delta$CGR) on the anomalies of climate variables over tropical land. We defined anomalies as the departure to the fitted trend line of a time series[8,20]. We used nine competing models (M1 to M9) to derive $\gamma_{CGR}^{T}$ and $\gamma_{CGR}^{W}$ for every 20-year moving window, alternatively using the anomalies of tropical mean annual temperature ($\Delta$MAT), mean annual precipitation ($\Delta$MAP), mean shortwave radiation($\Delta$RAD), 4-month lagged precipitation ($\Delta$MAP$_{lag}$), satellite-based terrestrial water storage ($\Delta$TWS) and the interactions of temperature and water proxies as the predictors (Table 1; Supplementary Fig. 1). The interaction term was added in some models to account for the interaction of water and temperature variability, as suggested in a recent study[15]. We evaluated the performance of models based on their time range covered, adjusted coefficient of determination (Adj. $R^2$), Akaike information criterion (AIC) and variance inflation factor (VIF). In particular, large VIF indicates collinearity in predictors, which can generate instability in the coefficients of linear models, leading to more uncertain climate sensitivities. Long-term dynamics of $\gamma_{CGR}^{T}$ and $\gamma_{CGR}^{W}$ were qualitatively similar between the different multivariate models tested (Supplementary Fig. 1). We selected M1 (precipitation based) and M3 (TWS based) for the analyses presented in the main text, as they were less influenced by predictor collinearity -- evidenced by their lower VIF scores. CGR in the years 1991–1993 were affected by the eruption of Mt. Pinatubo and thus excluded from the analysis.

**Quantifying drought-affected area.** We used the percentiles of local monthly precipitation to detect droughts and drought-affected area. Though multiple aridity index can be used to do so, we chose monthly precipitation because (1) the indicator is directly related to IPCC definition of drought-"prolonged absence or marked deficiency of precipitation[38]" (2) some known uncertainties in the estimation of water demand – potential evapotranspiration – in the derivation of aridity indexes[74,75]. Droughts occur when monthly precipitation is below a certain local threshold in a reference period[76]. Specifically, for each pixel in the tropics, we converted its monthly precipitation from 1959 to 2016 to a percentile distribution. We defined the months that belong to the bottom 1% percentile of the precipitation as experiencing very extreme drought. By changing the choice of precipitation threshold (e.g., 1, 5, 10, 15, 20, 25, 30, 40, and 50%), we identified droughts of different severity for the pixel—the bottom 1% precipitation is very extreme drought, 1–10% is extreme drought, 10–25% is mild drought, and 25–50% is natural water deficits. To get the long-term dynamics of drought-affected area, for each 20-year span, we obtained the frequency of extreme drought (i.e., the ratio of the number of months experiencing droughts to the total number of months in the 20-year span) at the pixel level. The frequency was then used to weigh the area of pixels as we aggregated the drought-affected area to regional scales (e.g., tropic forests, semi-arid, continents, pantropic).

**Quantifying the regional contributions to STD$_{CGR}$.** Based on the tight correlation between extreme drought-affected area and STD$_{CGR}$, we adopted a time-for-space substitution to quantify regional (i.e., tropical America, tropical Africa, tropical Asia, tropical semi-arid ecosystems and tropical forests) contributions to STD$_{CGR}$, using the drought-affected area detected for these regions. To consider the spatial variation of regions in their land-atmospheric $CO_2$ exchange capacity, we used the multiyear average FLUXCOM NEE map as the spatial weight, as FLUXCOM NEE represents our best estimate of the spatial variation in NEE[58] and the product shows potential in capturing the extreme drought influence on NEE (Fig. 2d; Supplementary Fig. 5).

We note that to estimate the regional contribution of extreme droughts to STD$_{CGR}$, it is necessary to account for all transient and long-term carbon fluxes incurred by extreme droughts. In theory, this can be simulated in process-based DGVMs. But current DGVMs, as our results have suggested (Fig. 2d), inadequately capture the drought impacts on STD$_{CGR}$. Another option is to use remotely sensed aboveground biomass (AGB)[77] rather than data-driven NEE as the spatial weight. However, the method would imply that extreme droughts induced carbon losses proportionally to AGB. The assumption was questionable as drylands have less biomass but usually comparable net carbon exchange than wet forests[13,14]. Therefore, using the data-driven net flux product (i.e., FLUXCOM NEE) as a weight would be the preferred available option to approximate the regional contribution to STD$_{CGR}$, as FLUXCOM NEE showed potential in capturing the extreme drought influence on NEE (Fig. 2d; Supplementary Fig. 5). Importantly, our study is designed to provide a first-order quantification of the regional contribution to STD$_{CGR}$ at coarse continental and biome scales, while the fine-scale variations at the pixel-level within each continent and biome remain to be addressed.

**Remove the influence of autocorrelation.** Autocorrelation is the correlation of a time series with a delayed copy of itself. Since we used a moving window to calculate STD$_{CGR}$, the value of STD$_{CGR}$ in a window is not independent from its counterparts in adjacent windows, leading to autocorrelation in the time series of STD$_{CGR}$. In addition, since we calculated long-term variability in tropical water availability (e.g., $\overline{MAP}$, $\overline{SWC}$) using the same moving window, it can also generate some autocorrelation in these time series. To remove the influence of autocorrelation:

1) First, we introduced the Durbin-Watson indicator (DW) to evaluate the autocorrelation of these times series. We estimated the lag 1 autocorrelation in the residuals ($e_i$) from the ordinary linear regression of the variables of interest (e.g., STD$_{CGR}$, $\overline{MAP}$) to time, where the coefficient in the regression is $\rho$, as in $e_i = \rho\, e_{i-1} + r_i$. The DW tests the null hypothesis that residuals are uncorrelated ($\rho = 0$), against the alternative hypothesis that autocorrelation exists ($\rho \neq 0$) (Eq. 1).

$$DW = \frac{\sum_{i=2}^{n}(e_i - e_{i-1})^2}{\sum_{i=1}^{n} e_i^2} \quad (1)$$

Where $n$ is the number of observations, $e_i$ is the $i$th residual of the linear regression of a target variable to time. The Durbin-Watson indicator (DW) is a value ranging between 0 and 4, where 0 means positive autocorrelation, 2 means no autocorrelation and 4 means negative autocorrelation.

2) We applied the Cochrane-Ocrutt procedure to adjust the variables of interests ($y_i$; e.g., STD$_{CGR}$, $\overline{MAP}$, $\overline{SWC}$, $\overline{TWS}$, $\overline{MAT}$, $\overline{VPD}$) to STD$_{CGR\_adj}$, $\overline{MAP_{adj}}$, $\overline{SWC_{adj}}$, $\overline{TWS_{adj}}$, $\overline{MAT_{adj}}$ and $\overline{VPD_{adj}}$ where $y_{i\_adj} = y_i - \rho y_{i-1}$. After the procedure, we found the autocorrelations in STD$_{CGR\_adj}$, $\overline{MAP_{adj}}$, $\overline{SWC_{adj}}$, $\overline{TWS_{adj}}$ and $\overline{MAT_{adj}}$ and $\overline{VPD_{adj}}$ were largely removed since their DW values were close to $2 - 1.6$, $1.3 \pm 0.1$, $1.2$, $1.3$, $1.9$ and $1.9 \pm 0.03$, respectively. Among them, $\overline{MAP_{adj}}$ and $\overline{TWS_{adj}}$ were negatively correlated to STD$_{CGR\_adj}$ with significance level of $p < 0.01$ and $p = 0.08$ (Fig. 2a), respectively, meaning the negative impact of long-term water availability on STD$_{CGR}$ we found is significant after considering autocorrelation.

Furthermore, we conducted an alternative test to remove autocorrelation, in which we divided the 58-year CGR records into twelve independent 5-year segments (only the last segment has 2-year overlap with its previous segment), and calculated $\gamma_{CGR}^T$, $\gamma_{CGR}^W$, STD$_{CGR}$ and the drought-affected area for each 5-year segment. Each 5-year segment is therefore independent from others since there is no overlap between them. Based on the $\gamma_{CGR}^T$, $\gamma_{CGR}^W$, STD$_{CGR}$ and the drought-affected area from these 5-year segments, we re-examined their relationships to support our conclusion (Supplementary Fig. 3c, d).

## Data availability

All data used in this study is publicly available. The Global Carbon Project dataset is archived at the website (https://www.icos-cp.eu/science-and-impact/global-carbon-budget/2017). The simulations from TRENDY DGVMs are available at https://sites.exeter.ac.uk/trendy. CRU TS4.01 can be accessed at https://crudata.uea.ac.uk/cru/data/hrg and CRU-NCEP can be accessed at https://crudata.uea.ac.uk/cru/data/ncep/. FLUXCOM dataset was downloaded from www.bgc-jena.mpg.de/geodb/projects/Data.php. Global Fire Emissions Database is accessible at http://www.globalfiredata.org/.

Berkeley Earth Surface Temperature can be downloaded from http://berkeleyearth.org/data/. Other auxiliary temperature and precipitation datasets are freely available at https://www.esrl.noaa.gov/psd/data/. We provide a processed dataset to support the reproduction and verification of the results at https://zenodo.org/record/5908612.

## Code availability

All code that supports the finding of this study is available at https://zenodo.org/record/5908612.

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

## Acknowledgements

X.L. and T.F.K. were supported by the NASA Terrestrial Ecology Program IDS Award NNH17AE86I. X.L. acknowledges support from National University of Singapore and Singapore Ministry of Education (A-0003625-00-00). T.F.K. also acknowledges additional NASA support through award #80NSSC21K1705, support by the Director, Office of Science, Office of Biological and Environmental Research of the US Department of Energy under Contract DE-AC02-05CH11231 as part of the RUBISCO SFA and a DOE ECRP Award DE-SC0021023. We acknowledge the researchers of the Global Carbon Project for making their data available. We thank the FLUXCOM group for the provision of upscaled terrestrial carbon fluxes. We thank Dr. Stephen Sitch, Dr. Pierre Friedling-stein, and all modelers who have contributed to the Trends in Net Land-Atmosphere Exchange project (TRENDY). We appreciate the auxiliary climate datasets provided by the NOAA/OAR/ESRL PSD, Boulder, Colorado, USA.

## Author contributions

X.L. and T.F.K. conceived the idea and designed the study. X.L. carried out the analysis. X.L. and T.F.K. wrote the paper.

## Competing interests

The authors declare no competing interests.

**Additional information**

