## [Peer Review File · Nature Communications]

Tropical extreme droughts drive long-term increase
in atmospheric CO₂ growth rate variabilityREVIEWER COMMENTS

Reviewer #1 (Remarks to the Author):

This manuscript attempts to reconcile the previous contradicting findings on the relative roles of tropical land temperatures vs water availability in shaping the inter-annual variability (IAV) of the terrestrial carbon balance. Specifically, it focuses on revealing the relationships among decadal changes (using a 20yr moving window) in IAV of CGR, temperature sensitivity, and climatologically mean water availability. This research theme is of interest to the audience of Nature Communication. However, I have several major concerns for the current analyses, conclusions drawn, and clarity/completeness/validity of approaches utilized.

1. The title – “Tropical drought frequency influences long-term dynamics of the temperature sensitivity of the global carbon cycle” is misleading and not really supported by the results presented in the manuscript. First, “Tropical drought frequency” (used in the title) is not equivalent to extreme drought area (metrics utilized in figures/results). Fig.1 investigated the relationship between temperature sensitivity and IAV of CGR (denoted as STD_CGR); Fig. 2 examined the relationship between STD_CGR and a) climatological mean of water availability and b) area impacted by extreme drought. None of these figures directly analyzed “drought frequency” or the relationship between “drought frequency” and “temperature sensitivity”. Given the current title, it would be much more straightforward to check the correlation between temperature sensitivity and drought frequency.

2. NEE was assumed by the authors to be equivalent to CGR, for both the IAV metrics (i.e., STD_NEE and STD_CGR in Fig. 2) and the temperature sensitivity metrics (i.e., gamma_NEE and gamma_CGR in Fig. 1), which is not actually the case. Fluxes from the Land use land cover (LULC) can play a significant role in modulating the IAV of CGR (Yue et al., 2020). Even though the authors checked the impact of LULC in Fig. S2, that LULC dataset was from the global carbon budget calculated with book-keeping models. Such models are forced with land-use reconstructions, and built upon fixed carbon densities and temporal response curves for different ecosystems following a land-use transition. The resulting LULC time series can be very smooth (Fig. 4 in Yue et al, 2020), dampening its contribution to the IAV of CGR.

Yue, C., Ciais, P., Houghton, R.A. et al. Contribution of land use to the interannual variability of the land carbon cycle. *Nat Commun* 11, 3170 (2020). <https://doi.org/10.1038/s41467-020-16953-8>

3. In Fig. 3 and line 199-202, “To further quantify the regional contributions of extreme droughts to global STDNEE, we applied the tight global relationship between extreme drought-affected area and STD_CGR (Figure 2c) to the extreme drought-affected area of each continent (i.e. tropical America, Africa and Asia) and biome (i.e. semi-201 arid/arid and rainforests), assuming the relationship holds at the continental and biome scales.” – This approach is very concerning, as “Extreme droughts happened unevenly over time and space, and thus contributed differently to the variation of STD_CGR” as the authors also acknowledged in the paper. The assumption stated above is simply unrealistic.

4. It seems that the authors assume extreme drought impacted area is equivalent to the climatological mean of water availability, which again is not the case. The two metrics differ significantly in terms of their statistical definition and timescales. Also, meteorological drought is ~ 3month, which defines a short-term drought and hence contrasts the multiple year lagged effect or long-term mean water availability stated in the paper.

5. Area impacted by extreme drought – is this biomass weighted? Given that “extreme droughts happened unevenly over time and space” as well as across biomes, and that different biomes have disproportionally contribution to CGR, simple area summation without biomass weighting does not seem to accurately and adequately characterize the resulting CGR changes due to droughts.

6. Description of datasets and methodology is either unclear or incomplete. A) NEE is defined for the global average or tropics only? B) No definition of tropical area. C) no description of how rainforests and semi-arid areas are defined, based on what dataset? D) where is the TWS dataset

from? The paper stated that it is satellite-based; yet satellite TWS from GRACE is only available since 2002 while this paper used TWS from 1980. E) statistics: both R^2 and r were used but no justification provided. F) uncertainty metrics: sometimes standard deviation was utilized but sometimes standard error was utilized; no justification was provided why going back and forth.

Given the major concerns above, I don't think the research aim is met – "reconcile the previous contradicting findings on the relative roles of tropical land temperatures vs water availability in shaping the inter-annual variability (IAV) of the terrestrial carbon balance".

Detailed comments:

Line 18-19: "The sensitivity of CGR to tropical temperature has changed markedly over the past six decades, however, an observation to date remains unexplained" – It should be the driver responsible for the decadal changes in temperature sensitivity not identified? Please be precise.

Line 22-24: The decadal changes in STD_CGR should not be the cause of three-fold increase in temperature sensitivity but rather a consequence. Therefore, the goal in line 18-19 does not seem to be met with the analyses conducted in the paper. Also, STD change by %? What is the reference to calculate %?

Line 25: how "tropical area" is defined? Be clearer on what biomes are included.

Line 26-27: "A 1% increase in the tropical area affected by extreme droughts led to about 0.14 Pg C yr⁻¹ increase in STDCGR" - this really depends on the location and timing of drought events, which can be highly heterogeneous in space, time. It also depends on biomes that are struck by droughts, as different biomes have different biomass.

Line 27-28: "a few extreme droughts influence STD_CGR, which amplifies temperature sensitivity" – not really a few extreme droughts; how come drought occurrences can amplify temperature sensitivity?

Line 29: "long-term variability in CGR" – more precisely, it should be long-term changes in CGR variability?

Line 34: "relatively small contribution from land use emissions" – see Yue et al., 2020 above.

Line 24-35: "deltaCGR highly variable" – this is variability of variability? How come delta CGR is negative? Should be CGR?

Line 36: "tropical climate" – this is vague, tropics rainforests and tropical semi-arid regions have very different climate?

Line 36: "through the influence of climate on the tropical terrestrial carbon sink" – this may not be always the case. For example, fire can overwhelm climate impact in Southeast Asia (Liu et al., 2017).

Liu et al., 2017. Contrasting carbon cycle responses of the tropical continents to the 2015–2016 El Niño. *Science*.

Line 40: "as have drier years" – odd phrase.

Line 48: replace "than" with "from"

Line 55: "due to the largest aboveground biomass (ref 27)" – this is not clear, without distinction between rainforest and semi-arid biomes.

Line 59: "long-term changes in frequency of extreme drought" – not really about frequency.

Line 60: "We examine relationships of γ_{CGR_T} and γ_{CGR_W} with tropical droughts" – this is not

really true. See my comments above.

Line 69-70: satellite-derived terrestrial water storage – need to provide more info. What satellite provides such info since 1980?

Line 76: “decreasing by 33.6%” - how such % was calculated? Which period it is for the denominator.

Line 78: $p > 0.05$. what if a p value = 0.1? is the result sensitive to the p value cutoff?

Line 78: replace “negligible” role – insignificant. Also “suggesting a negligible role of tropical water availability in controlling the interannual variability of CGR.” – contradicting with conclusions made later.

Fig. 1a: add which statistical model is used for each curve in the caption.

Line 89: replace “non-significant” to “insignificant”

Line 90: 1000 bootstrap estimates: per 20yr window? no info at all for how bootstrap simulations were conducted.

Line 91: uncertainty quantification: standard error vs standard deviation. CI.

Line 93: first value? Value in the first year in each 20yr window?

Line 99-100: “We found that both DGVMs and the machine learning products mischaracterized the 99 temperature sensitivity (γ_{NEET}) of terrestrial net ecosystem exchange (NEE)” – not surprising, but how about LULC? – see my comments above.

Line 104-108: Nice to see this clear statement! Fully agree.

Line 115-116: “we found that the STD_CGR increased 34.8% from the 1960s to the 2000s, and then decrease slightly by 13.2% in the recent decade” – what is the denominator in calculating such % changes?

Line 116-117: such a change in STD_CGR explained 65% of the variance in and such a change in temperature sensitivity. – again, I don’t think STD_CGR is the cause.

Line 119-120 and 440: LULC changes from Global Carbon Budget, which may have a very smoothed CGR. See my comments above.

Line 123-124: “The long-term dynamics of STD_CGR over the past 60 years are independent of the year-to-year climate variability but driven by a long-term factor.” – the claim seems very odd, and even contradicting with previous statements.

Line 125: “long-term water availability” - can be confusing. What is long-term?

Line 138: “long-term water availability” should be climatological mean?

Line 141: uncertainty: 1 standard error again

Line 142: area affected by droughts? What is the denominator? – global area?

Line 143-144: Percentile of precipitation – to establish the long-term statistics, using each 20yr moving window, or across the entire period?

Line 147: why both linear regressions with and without intercept were examined?

Line 149: The relationship in panel C is sensitive to the cutoff value used in the definition of extreme drought, as shown in panel d. But this was interpreted as mild drought has a limited influence on STD_CGR (line 178-179). Also, When drought intensity is low, is the correlation of STD_CGR and area affected by extreme drought still significant? Why suddenly r not R^2 ?

Line 161: Meteorological drought (line 161) is short term, in contrast to the multiple year lagged effect or long-term mean TWS stated above

Line 164: drought-affected area should be biomass weighted.

Line 166: drought duration – very different from drought frequency defined in the title. No information provided in the method section

Line 166-170: percentage change, what is the denominator, given the different time period?

Line 170-171: “the long-term change in tropical water availability was primarily driven by the increasing occurrence of extreme droughts.” – this does not sound logical, and the previous statements do not really directly support this claim.

Line 176-178: “outsize influence” – really depends on where the drought occurs, as later acknowledged by authors – “extreme droughts happened unevenly over time and space, and thus contributed differently to the variation of STD_CGR”

Line 188-189: when comparing tropical Africa and tropical America and Asia, the extreme impacted drought area was calculated using the total pan tropical area (which is consistent across three regions) or the respective total tropical area in each continent? Also, is the relationship found in Fig. 2-3 sensitive to the definition of tropical area, which was never clearly defined?

Line 189: tropical America Africa?

Line 199-202: this does not make sense, see my comments above

Line 203: how semi-arid/arid ecosystems vs rainforests were defined? What datasets were used?

Line 205-209: cannot really compare across continents IF using the scaling approach defined in line 199-202. Very different biomes. Even rainforests are highly different between Amazon and Congo.

Line 218: The extreme-induced STD_NEE (Fig. 3b, d, f) were estimated using the relationship between drought-affected area and STD_CGR (Fig. 2c). STD_NEE and STD_CGR totally different things and were never validated. How to interpret the differences in Pan Tropic and CGR?

Line 225-226: STD_CGR highly correlated with temperature, and also highly correlated with decadal scale mean TWS. But what is the cause, what is the consequence, and what is the constraint? Here everything is based on correlational analysis.

Line 226-227: we attribute the strong coupling between STD_CGR and long-term water availability to the effects of extreme drought on tropical terrestrial carbon cycling. But extreme drought is not equivalent to long-term water availability. See my comments above.

Line 236: our results help reconcile conflicting reports on the water sensitivity of CGR – not sure if this goal is met.

Line 241-242: “It is possible that extreme climate events – which are likely underrepresented by the relatively short and sparse eddy covariance measurements – led to structural and long-term changes that are undetected by flux sites” - not sure if this claim is really true. Most of the EC measurements so far are longer than 20yr window. I think the problem in FLUXCOM is more related to the upscaling approach, which cannot really conserve the temporal variability.

Line 255-256: our results show that extreme droughts also influence CGR at longer time scales by amplifying STD_CGR and temperature sensitivity. How so? This paper does not really examine this amplification?

Line 290-291: Our analysis shows that long-term tropical water availability is negatively related to the changes in STD_CGR, which explains the CGR pronounced variations in temperature sensitivity, as CGR is more variable when there are more extreme droughts. – this claim is self-contradicted, and also contradicted to Fig. 2a

Line 318: monthly soil water content: uncertainty of SWC calculated from the SPLASH model?

Line 359: satellite-based TWS: no further info. where is this from? GRACE is only available since 2002

Reviewer #2 (Remarks to the Author):

Summary:

This is a review of the manuscript Tropical drought frequency influences long-term dynamics of the temperature sensitivity of the global carbon cycle (NCOMMS-21-08853) submitted to Nature Communications. This manuscript is well suited to the topics of Nature Communications as it is an investigation of why the sensitivity of atmospheric CO₂ growth rate (CGR) to tropical temperatures – already established in literature – changes over time. The manuscript uses observations of atmospheric CO₂ concentration along with observations/reanalysis of precipitation, temperature, radiation, and water availability to calculate the sensitivity of CGR to tropical temperatures using multiple model formulations for a detailed model selection. From this effort they show that the models are generally consistent in their retrieval of the sensitivity of CGR interannual variation with Temperature and water (precipitation or storage). They perform these regressions along a moving 20 year window to show that the sensitivity of CGR to Temperature has changed over time, while the (relatively low) sensitivity of CGR to Water has not. Additionally, these results were compared to similar results (using net ecosystem emissions rather than CO₂ concentrations) from dynamic vegetation models and FluxCom.

They then correlated the change in the sensitivity of CGR to a change in CGR standard deviation along with a long term change in water storage and the extent of extreme drought. Their conclusion is that the change in land area impacted by extreme drought drove a large part of the increased sensitivity of CGR to temperature, particularly in Tropical Africa.

I found this to be a well written manuscript on an important topic relating to how the tropical biosphere function will respond to long term change. However, I have reviewed a previous version of this manuscript and continue to be concerned by reporting r-squared and significance statistics based on running regressions in the main manuscript. Each regression coefficient calculated in a running regression includes a large amount of non-independent data. Accounting for this will likely significantly – and I would suggest appropriately – reduce the statistical power of this analysis.

Though this is a major concern, I do think that the inference in this manuscript may still be supported and will benefit from a more conservative analysis. These concerns relate to both the analysis presented in Figure 1d (relating standard deviation of CGR and sensitivity of CGR to Temperature) and Figure 2a and 2c (relating area of extreme drought to change in sensitivity of CGR to Temperature).

I suggest a major revision for the calculation of the change of sensitivity over time to account for the autocorrelation inherent in a running calculation or do away with it entirely in supporting the inference of the manuscript. Other minor revisions detailed below.

Reviews of [figures] and [lines] below.

Minor revisions:

[189] Possible typo, correct for clarity: 'tropical America Africa'

[241 - 244] When discussing FluxCom and the lack of sensitivity, it should be noted that FluxCom interannual variability is also reduced compared to the flux towers that underpin the product (Tramontana et al. 2016). Additionally, there are very few flux towers available or included in FluxCom in the tropics.

[257 – 261] This discussion of mechanisms would benefit from including discussion of vapor pressure deficits direct impact on gross primary productivity and its non-linear response to temperature. Increased Temperatures in the tropics should lead to increased variability of VPD even if the variability of Temperature stays the same.

[Figure 1d] The relationship between the standard deviation of CGR with the Temperature simply due to the relative correlation of Temperature of with CGR – such that any increase in standard deviation will result in an increase in sensitivity (being the combination of correlation and the ratio of standard deviations of Temperature and CGR). Does the correlation between Temperature and CGR or the standard deviation of Temperature change at all through the time period?

References

Tramontana, G. et al. Predicting carbon dioxide and energy fluxes across global FLUXNET sites with regression algorithms. *Biogeosciences* 13, 4291–4313 (2016).

Reviewer #3 (Remarks to the Author):

This paper links variability in the rate of growth of atmospheric CO₂ to variability in tropical land surface temperature and assorted metrics of tropical land surface water availability. The authors find that variability in atmosphere CO₂ growth rate (CGR) increased over much of latter 20th century, but has recently decreased. The sensitivity of CGR to tropic temperature likewise increased during the late 20th century, but decreased during the early 21st century. The authors report that the sensitivity of CGR to tropical water availability is weak, but that the area of tropical land affected by extreme drought is strongly correlated with variability in CCR.

Overall, I found the methods appeared to be state of the art, and in general I was able to understand clearly what the authors did (but see a few notes on presentation below). I think the paper is interesting.

I offer the following generally notes and reflections:

First, I think the study would benefit from clearer attention to context, especially in terms of what's been done already on the topic. This is not the first study seeking to link variability in CGR or land carbon uptake to temperature in the tropics; the authors mention that their study is preceded by earlier work, but it would be nice if they could offer a summary of what has already been learned from the work focused on tropical climate, and also the complimentary literature suggesting it is actually dryland C uptake sensitivity that drives interannual variability in the global carbon sink anomalies (e.g. Poulter et al. 2014). Then, the authors could identify the specific knowledge gaps that remain (i.e. identify the questions that will be answered in this study), and also discuss who benefits from the knowledge that will be gained? It seems the primary audience is the modeling community...are there other groups of scientists or stakeholders who should be interested specifically in the links between variability in CGR (as opposed to the mean trend) and tropical climate?

There seem to be a substantial number of scientists in the field who are seeking for "silver bullet" explanations for variability in the land carbon sink, and that this has been going on quite some time. But the problem is that results from one study to the next are often contradictory, except in their conclusion that the "models are wrong." So basically, I'm looking for a clear explanation of

why we need another study that comes to the same conclusion right now.

Next, I have two observations about methodology and interpretation. First, when specifically considering the relationship between “extreme drought” and CGR variability, what the authors have shown is correlation, but not causation. Is it possible that the occurrence of extreme drought in the tropics is itself driven by other mechanisms (e.g. El Nino or other global scale phenomena) which might affect the land carbon sink in many ways, not just by making drought more likely in the tropics? The strong correlation between the area of tropical land under extreme drought and the std(GCR) (Fig 2c) is certainly striking. But it is really possible that this small area of land could have such an outsized influence on the global land carbon sink? It seems that, at a minimum, some back-of-the-envelope calculations based on our best guesses of NEE in the tropics, and its sensitivity to drought stress, would be informative. For example, consider a hypothetical case where one year, none of the tropical land area is under drought stress, and another where let’s say 8% of the land area is under drought stress. What is the expected change in the tropical NEE, and what fraction of global NEE does that change represent? Are the results coherent with the slope of the line appearing in Fig 2c?

Speaking of linear slopes, that brings me to my second comment about methodology. I felt that the determination of the sensitivities of CGR to temperature and water availability using multiple linear regressions (e.g. Table 1) was overly empirical. Is a simple linear regression really the best way to approximate processes that are known to be highly non-linear? Consider, for example, the case of water storage, which we know will have a threshold-type relationship with NEE...when soil is well-watered, we would expect a priori that NEE should be insensitive to variability in soil water; on the other hand, when it’s dry, the relationship between the two will become more pronounced. Should we really be drawing a line through data that represent a threshold-driven process? Likewise, with respect to temperature, again...is a linear relationship expected? Or perhaps it’s more exponential, reflecting the exponential relationship between respiration and temperature, compounded by reductions in photosynthesis when T is high. In any event, I admit it’s hard for me to understand why results of a simple linear regression should be interpretative as informative of the apparent “failure” of more complex and mechanistic earth system models.

Finally, I was surprised to see so much emphasis on drought legacy effects in the discussion, as this wasn’t really a focus of the analysis. On lines 254-257, the authors state “Several studies have suggested that a few extreme events explained a significant amount of the variance in land-atmosphere carbon exchange, at seasonal or interannual time scales. However, our results show that extreme droughts also influence CGR at longer time scales by amplifying STD_CGR.” Maybe I’m missing something, but I don’t see how a relationship between extreme droughts and STD_CGR necessarily implies an important role for legacy effects. This area of the discussion would benefit from a review of independent evidence, even from site level studies, of the extent to which drought legacies characterize the dynamics of carbon exchange in the tropics. While I am not an expert on this topic, I would think that the biodiversity and mesic conditions that characterize much of the region should confer a fair amount of resilience to the impacts of extreme droughts.

A few more minor comments:

Abstract: Would it be possible to write an abstract that does not rely so strongly on symbols and abbreviations? It took me a long time to get through it, as I had to first learn what the different symbols referred to. A plain language description would seem more appropriate.

As I’m sure the authors know, a study was published in Nature (Humphrey et al. 2021) recently that seems very relevant to the present study...in particular, the paragraph of Humphrey et al that contains the sentence: “Thus, although the IAV in global land carbon uptake has been empirically found to be sensitive to tropical mean temperature in numerous studies, our results suggest that this sensitivity does not represent a strong mechanistic link...” The authors of this study will probably need to respond accordingly.

The decrease in CGR variability over the past two decades is really interesting, and seems like a good source of novelty for this study when compared to similar studies conducted before this

declining trend became so apparent. The authors might consider emphasizing this aspect of their results more strongly. That the shift in slope occurs is mentioned in the abstract, but the reasons for it are not discussed.

The authors spend a fair amount of ink describing how their results highlight model deficiencies. But what about the FLUXCOM product? Any recommendations for how that might be improved?

References:

Humphrey, V., Berg, A., Ciais, P., Gentile, P., Jung, M., Reichstein, M., Seneviratne, S.I. and Frankenberg, C., 2021. Soil moisture–atmosphere feedback dominates land carbon uptake variability. *Nature*, 592(7852), pp.65-69.

Poulter, B., Frank, D., Ciais, P., Myneni, R.B., Andela, N., Bi, J., Broquet, G., Canadell, J.G., Chevallier, F., Liu, Y.Y. and Running, S.W., 2014. Contribution of semi-arid ecosystems to interannual variability of the global carbon cycle. *Nature*, 509(7502), pp.600-603.

Tropical extreme droughts associated with long-term increase in global carbon cycle variability

NCOMMS-21-08853 Response to Reviewers

Authors: We appreciate the constructive comments from the reviewers and the invitation from the editor to submit a revised version. We have carefully followed the reviewers' suggestions to clarify our methods, conduct additional analyses and enhance the discussion. Please see our point-to-point response below in blue text.

We have made the following major changes:

1. Highlight that our objective is to examine the long-term characteristic of atmospheric CO₂ growth rate (i.e., STD_{CGR}), not the CGR change at the interannual time scale.
2. Update key correlation analysis using the autocorrelation corrected time series.
3. Add a full description of ancillary data, including a new land use emission estimate.
4. Use FLUXCOM NEE as a spatial weight to improve the quantification of regional contributions to the changes in STD_{CGR} .
5. Enhance the discussions on how to improve DGVMs and FLUXCOM for the simulation of long-term carbon cycle dynamics.
6. Add a new discussion on the potential non-linearity of climate sensitivities and its implications for our study.

Reviewer #1 (Remarks to the Author):

R1C1: This manuscript attempts to reconcile the previous contradicting findings on the relative roles of tropical land temperatures vs water availability in shaping the inter-annual variability (IAV) of the terrestrial carbon balance. Specifically, it focuses on revealing the relationships among decadal changes (using a 20yr moving window) in IAV of CGR, temperature sensitivity, and climatologically mean water availability. This research theme is of interest to the audience of Nature Communication. However, I have several major concerns for the current analyses, conclusions drawn, and clarity/completeness/validity of approaches utilized.

Authors: Thank you for the meticulous and constructive comments. We have endeavored to take your comments onboard to improve our manuscript.

R1C2: 1. The title – “Tropical drought frequency influences long-term dynamics of the temperature sensitivity of the global carbon cycle” is misleading and not really supported by the results presented in the manuscript. First, “Tropical drought frequency” (used in the title) is not equivalent to extreme drought area (metrics utilized in figures/results). Fig.1 investigated the relationship between temperature sensitivity and IAV of CGR (denoted as STD_{CGR}); Fig. 2 examined the relationship between STD_{CGR} and a) climatological mean of water availability and b) area impacted by extreme drought. None of these figures directly analyzed “drought frequency” or the relationship between “drought frequency” and “temperature sensitivity”. Given the current title, it would be much more straightforward to check the correlation between temperature sensitivity and drought frequency.

Authors: We apologize for the confusion. We agree with the reviewer that the title should more strictly reflect our objective and results. Therefore, we have revised the title to: “Tropical extreme droughts associated with long-term increase in global carbon cycle variability”.

R1C3: 2. NEE was assumed by the authors to be equivalent to CGR, for both the IAV metrics (i.e., STD_{NEE} and STD_{CGR} in Fig. 2) and the temperature sensitivity metrics (i.e., γ_{NEE} and γ_{CGR} in Fig. 1), which is not actually the case. Fluxes from the Land use land cover (LULC) can play a significant role in modulating the IAV of CGR (Yue et al., 2020). Even though the authors checked the impact of LULC in Fig. S2, that LULC dataset was from the global carbon budget calculated with book-keeping models. Such models are forced with land-use reconstructions, and built upon fixed carbon densities and temporal response curves for different ecosystems following a land-use transition. The resulting LULC time series can be very smooth (Fig. 4 in Yue et al, 2020), dampening its contribution to the IAV of CGR.

Yue, C., Ciais, P., Houghton, R.A. et al. Contribution of land use to the interannual variability of the land carbon cycle. Nat Commun 11, 3170 (2020). <https://doi.org/10.1038/s41467-020-16953-8>

Authors: Thank you for pointing this out. We agree with the reviewer that land use can potentially influence the interannual variability (IAV) of CGR. We have indeed assessed the impacts of land use and land cover change (LULC) on the long-term dynamic of STD_{CGR} using the results from two bookkeeping models (Le Quéré et al., 2018). We have to clarify that our study is examining the change in long-term CGR variability (i.e., indicated by the standard deviation of CGR over 20-year window; STD_{CGR}) not the CGR interannual variability. Though LULC may have considerable contribution to CGR IAV (Yue et al., 2020), it does not necessarily explain the change in STD_{CGR} in the past six decades.

Following the suggestion by the reviewer, we acquired the modelled LULC emissions from Yue et al. 2020 (Yue et al., 2020) and added the magnitude of LULC emissions IAV in Fig. S2a (named as CY-LULC). The updated results suggest that although the LULC emissions from Yue et al. 2020 have a larger variability than the bookkeeping results (i.e., larger STD_{flux}), they show different temporal dynamics than that of STD_{CGR} (the black line in Fig. S2a). This suggests that LULC emissions are not responsible for the STD_{CGR} changes we examine.

Figure S2. Temporal changes in the magnitude of the interannual variability of CGR, ocean uptake, land use change emissions, fire emissions, leaf area index (LAI) and climate variables. Interannual variability is indicated by one standard deviation of annual values (STD) within 20-year moving window. (a) CGR, ocean carbon uptake and land use change emissions are provided by the Global Carbon Budget. CY-LULC indicates the land use change emissions estimated by ORCHIDEE (Yue et al., 2020), which have considered the changes in carbon density and climate variability for the land use change emissions. Fire emissions are obtained from Global Fire Emissions Database (GFEDv4.1). Since fire emissions record only range from 1997 to 2020, we used 10-year moving window to calculate the STD of fire fluxes. The shadings of ocean uptake indicate one standard error of eight models. LAI is the GMMIS AVHRR3g LAI over the tropics from 1982 to 2016. (b) STD of air temperature (Tair), precipitation (Precip), vapor pressure deficit (VPD) and soil water content (SWC) are normalised by the STD of first 20-year window (1959-1979) to make them comparable.

R1C4: 3. In Fig. 3 and line 199-202, “To further quantify the regional contributions of extreme droughts to global STD_{NEE} , we applied the tight global relationship between extreme drought-affected area and STD_{CGR} (Figure 2c) to the extreme drought-affected area of each continent (i.e. tropical America, Africa and Asia) and biome (i.e. semi-201 arid/arid and rainforests), assuming the relationship holds at the continental and biome scales.” – This approach is very concerning, as “Extreme droughts happened unevenly over time and space, and thus contributed differently to the variation of STD_{CGR} ” as the authors also acknowledged in the paper. The assumption stated above is simply unrealistic.

Authors: Thank you for the comment. We agree with the reviewer that a more sophisticated consideration of spatial and temporal variation is desired to accurately assess the regional contribution to STD_{CGR} . Motivated by this comment and R1C6 from the reviewer, we now use a data-driven net ecosystem exchange (FLUXCOM NEE) map as a spatial weight and updated the regional contributions to STD_{CGR} . FLUXCOM NEE represents our best estimate of the spatial variation in NEE (Jung et al., 2020) and it has been found to capture the extreme drought effect to some degree (Fig. 2d; Fig. S5). We have updated Figure 3 and its associated numbers in the text based on this new approach, and we have added a new section in the Methods to describe the procure in details (L446-452).

Figure 3. Spatial distribution of extreme droughts over the past 60 years. (a) the fraction of tropical vegetated area affected by extreme droughts (20-year moving average) for tropical (T.) forests and semi-arid regions, the shaded area indicates interannual variability in the fraction (e.g., one standard deviation); (c) the fraction of tropical vegetated area affected by extreme droughts for tropical America, tropical Africa and tropical Asia; (e) the changes of the extreme drought hotspots in three 20 year periods. Hotspots are defined as regions that are under extreme droughts for more than 10% of time; (b, d, f) The impacts of tropical extreme droughts on the interannual variability of global net ecosystem exchange (STD_{NEE}) for each 20-year period. The extreme-induced STD_{NEE} was estimated using the relationship between drought-affected area and STD_{CGR} (Fig. 2c) in combination with a spatial

weight based on FLUXCOM NEE (see Methods). The average impact from each region is indicated by the numbers in the figure, while the error bars indicate interannual variability of the impact. The impact of droughts estimated by DGVMs and FLUXCOM are included as reference, where the error bars indicate the inter-model variation. Fire emissions are obtained from Global Fire Emissions Database (GFED4s).

R1C5: 4. It seems that the authors assume extreme drought impacted area is equivalent to the climatological mean of water availability, which again is not the case. The two metrics differ significantly in terms of their statistical definition and timescales. Also, meteorological drought is ~ 3 month, which defines a short-term drought and hence contrasts the multiple year lagged effect or long-term mean water availability stated in the paper.

Authors: Sorry for the lack of clarity regarding this point. We did not assume extreme drought impacted area is equivalent to the climatological mean of water availability, and fully agree with the reviewer that that would not be a well-founded approach. We instead used the localized percentile of monthly precipitation to identify drought events. Specifically, for each 0.5 degree pixel in the tropics, we convert its precipitation to a percentile distribution. We defined the months that experiencing the bottom 1% percentile of precipitation as experiencing very extreme drought. We defined the months that experience the bottom 10% local precipitation as extreme droughts. We repeat the process for each pixel and for each threshold (i.e., 1%, 5%, 10%...) to get the durations of different magnitudes of droughts for all tropical vegetated land pixels.

Then for long-term dynamics (at the time scale of 20 years), we used the concept of drought frequency, which is the ratio of the number of months (n) that is classified as extreme drought to the total number of months ($n/20*12$). By doing this, we obtained the likelihood of extreme drought for each pixel in any 20 years span. The frequency was used to weigh drought-affected area when we aggregate the drought-affected area to regional scales (e.g., tropic forests, semi-arid, continents, pan tropic). We have revised the relevant parts in the Methods to clarify the approach taken (L429-444).

Following the suggestion from the reviewer, we tested an alternative case of using 3-month average precipitation to detect meteorological droughts, instead of the 1-month precipitation we used in the main analysis. We found that doing so did not affect our results (Fig. R1 vs Fig. 2c and d), as the correlation between drought area and STD_{CGR} is still significant ($p < 0.01$). The limited difference among the two cases is likely due to the fact we are examining long-term variability (20-year) in drought area, and their differences are averaged out when applying the 20-year window for drought frequency estimation.

Figure R1. The relationships between the changes in STD_{CGR} and tropical water availability and drought-affected area, with drought affected area calculated based on 3-month average precipitation. (a) The relationship between STD_{CGR} and the tropical area affected by extreme and very extreme droughts. The shading indicates the 95% confidence intervals of the linear regressions. The linear regressions with and without a y intercept were examined; (b) the correlation coefficient (r) between observed STD_{CGR} or modelled (i.e., DGVMs and FLUXCOM) STD_{NEE} and areas affected by droughts of different intensities in the tropics. Drought intensities are defined by the bottom percentiles (e.g., 1%, 10%, 25%) of 3-month average precipitation. Shaded areas indicate the inter-model variation of r (i.e., one standard error).

R1C6: 5. Area impacted by extreme drought – is this biomass weighted? Given that “extreme droughts happened unevenly over time and space” as well as across biomes, and that different biomes have disproportionally contribution to CGR, simple area summation without biomass weighting does not seem to accurately and adequately characterize the resulting CGR changes due to droughts.

Authors: Thank you for the suggestion. Please see our response to R1C4 in which we used the data-driven flux from FLUXCOM as a spatial weight in quantifying regional contributions to STD_{CGR} .

R1C7: 6. Description of datasets and methodology is either unclear or incomplete. A) NEE is defined for the global average or tropics only? B) No definition of tropical area. C) no description of how rainforests and semi-arid areas are defined, based on what dataset? D) where is the TWS dataset from? The paper stated that it is satellite-based; yet satellite TWS from GRACE is only available since 2002 while this paper used TWS from 1980. E) statistics: both R^2 and r were used but no justification provided. F) uncertainty metrics: sometimes standard deviation was utilized but sometimes standard error was utilized; no justification was provided why going back and forth.

Authors: We apologize for the insufficient details provided. We agree that these are very important and have made every effort in this revision to clarify each of these points. Specifically, we have made the following additions in the methods section (L369-380): (A) and (B). We defined tropics as the vegetated land surface between 23°S to 23°N, following the definition from Wang et al. (Wang et al., 2014). The NEE is the global NEE, which is directly comparable with CGR. (C). The land cover map was acquired from the MODIS MOD12 land cover product (Friedl et al., 2010). We have added the description of the data in a new methods section. (D). We used a reconstructed TWS dataset which ranges from 1981 to 2017 (Humphrey et al., 2018). (E). We have revised the manuscript to use r^2 where possible to indicate the tightness of correlations, but used r on one occasion (Fig. 2d) to show the direction of the correlation. (F). We have revised the paper to use standard error throughout the manuscript where appropriate.

R1C8: Given the major concerns above, I don't think the research aim is met – “reconcile the previous contradicting findings on the relative roles of tropical land temperatures vs water availability in shaping the inter-annual variability (IAV) of the terrestrial carbon balance”.

Authors: We apologize for the insufficient details and thank you for your suggestions for improvement. We have removed the statement “reconcile the previous contradicting findings...” from our list of objectives. Our objective is to provide an explanation for the observed long-term change in CGR variability (i.e., STD_{CGR}).

Detailed comments:

R1C9: Line 18-19: “The sensitivity of CGR to tropical temperature has changed markedly over the past six decades, however, an observation to date remains unexplained” – It should be the driver responsible for the decadal changes in temperature sensitivity not identified? Please be precise.

Authors: We have revised the statement as suggested.

R1C10: Line 22-24: The decadal changes in STD_{CGR} should not be the cause of three-fold increase in temperature sensitivity but rather a consequence. Therefore, the goal in line 18-19 does not seem to be met with the analyses conducted in the paper. Also, STD change by %? What is the reference to calculate %?

Authors: Thank you for this comment. While we agree with the reviewer that it is logical to say that the “true” climate sensitivity drives the changes in STD_{CGR} , the derivation of “true” climate sensitivities is challenging and highly debated, as we show in Figure S1 that the pattern and

value of climate sensitivity are dependent on method and data sources, and the assumptions of the linear models.

Meanwhile, we know that the changes in temperature sensitivity are, mathematically, indicative of the variability of CGR and the variability of temperature. While the variability of temperature (i.e., Fig. S2b) did not explain the large changes in the temperature sensitivity in the past six decades, we note that the variability of CGR, which has changed dramatically, could explain most of the change in the apparent temperature sensitivities (i.e., Fig. 1d).

Additionally, CGR is the least uncertain term the global carbon cycle. As such we believe a study directly on STD_{CGR} is more robust than examining derived climate sensitivities. We have added the discussion above in L68-71.

We have revised the statement to say "... which increased 34.8% from 1960-1979 to 1990-2009 and subsequently decreased 13.2% in 1997-2016." to clarify the time period of comparison.

R1C11: Line 25: how "tropical area" is defined? Be clearer on what biomes are included.

Authors: Sorry for the lack of clarity. we have revised it to "tropical vegetated area (23°S – 23°N)", following the definition in Wang et al. (Wang et al., 2014). The vegetated area would include forests and semi-arid grassland and savannah ecosystems (L372-375).

R1C12: Line 26-27: "A 1% increase in the tropical area affected by extreme droughts led to about 0.14 Pg C yr⁻¹ increase in STD_{CGR} " - this really depends on the location and timing of drought events, which can be highly heterogeneous in space, time. It also depends on biomes that are struck by droughts, as different biomes have different biomass.

Authors: Thank you for the comment. The relationship reported here is for the pan tropics, meaning a 1% increase in extreme drought-affected area (denominator: pan tropical vegetated surface) in the tropics can lead to 0.14 Pg C yr⁻¹ increase in STD_{CGR} . Please see our response to R1C4 where we used FLUXCOM NEE as a spatial weight to better consider the spatial contributions to the changes in STD_{CGR} .

R1C13: Line 27-28: "a few extreme droughts influence STD_{CGR} , which amplifies temperature sensitivity" – not really a few extreme droughts; how come drought occurrences can amplify temperature sensitivity?

Authors: We have removed "a few" as it may cause confusion. We have rephrased it to "the outsized influence of extreme droughts on tropical vegetated surface amplifies...". Our results suggest that extreme drought-affected area is highly correlated with STD_{CGR} , which in turn explains the observed increase in the apparent temperature sensitivity.

R1C14: Line 29: “long-term variability in CGR” – more precisely, it should be long-term changes in CGR variability?

Authors: Thank you. We have removed the statement in this revision.

R1C15: Line 34: “relatively small contribution from land use emissions” – see Yue et al., 2020 above.

Authors: We have cited the Yue et al. 2020 paper here by saying “but see (Yue et al. 2020)”.

We acknowledge that the Yue et al. paper suggests a large contribution of LULC to IAV, but we have reservations about their conclusions, in that it is possible that climate change influences are classified as LULC change in this model-based study. We therefore prefer to use the more widely accepted LULC emissions in our main analyses but acknowledge that the Yue paper presents different conclusions. We also assessed the implications of using the Yue et al. LULC carbon emissions (Fig. S2a) and show it does not explain the long-term changes in CGR variability (Please kindly refer also to our response to R1C3).

R1C16: Line 24-35: “deltaCGR highly variable” – this is variability of variability? How come delta CGR is negative? Should be CGR?

Authors: Δ CGR is the detrended CGR (atmospheric CO₂ growth rate), meaning annual CGR anomalies. We have removed the trend in CGR to get the interannual variability in CGR, following the methods in Wang et al 2014 (Wang et al., 2014), which can lead to negative Δ CGR values.

R1C17: Line 36: “tropical climate” – this is vague, tropics rainforests and tropical semi-arid regions have very different climate?

Authors: We have updated the statement to say “...through the influence of climate on tropical terrestrial ecosystems(Bousquet et al., 2000; Fan et al., 2019; Peylin et al., 2013), among which tropical forests(Cox et al., 2013) or semi-arid ecosystems (Ahlström et al., 2015; Poulter et al., 2014) or both (Piao et al., 2020) have been reported to play the primary role.”

R1C18: Line 36: “through the influence of climate on the tropical terrestrial carbon sink” – this may not be always the case. For example, fire can overwhelm climate impact in Southeast Asia (Liu et al., 2017).

Liu et al., 2017. Contrasting carbon cycle responses of the tropical continents to the 2015–2016 El Niño. Science.

Authors: We have examined the long-term role of fire using the GEDs4 fire emission dataset (https://daac.ornl.gov/VEGETATION/guides/fire_emissions_v4.html). We agree that some cases of fire can make strong contribution CGR IAV (Van Der Werf et al., 2017), though current evidence does not support a role of long-term changes in fire emissions dominating long-term changes in STD_{CGR} (Fig. S2a).

R1C19: Line 40: “as have drier years” – odd phrase.

Authors: The phrase indicates that drier years could also cause anomalies in photosynthesis and respiration, similar to positive temperature anomalies.

R1C20: Line 48: replace “than” with “from”

Authors: We have revised this as suggested.

R1C21: Line 55: “due to the largest aboveground biomass (ref 27)” – this is not clear, without distinction between rainforest and semi-arid biomes.

Authors: We have removed the statement to make the logic flow tighter.

R1C22: Line 59: “long-term changes in frequency of extreme drought” – not really about frequency.

Authors: We have removed this as suggested.

R1C23: Line 60: “We examine relationships of γ_{CGR_T} and γ_{CGR_W} with tropical droughts” – this is not really true. See my comments above.

Authors: This has been removed as suggested.

R1C24: Line 69-70: satellite-derived terrestrial water storage – need to provide more info. What satellite provides such info since 1980?

Authors: We have rephrased it to “reconstructed satellite TWS”. We have added a new methods section to describe the dataset in detail. Please see our response to R1C7.

R1C25: Line 76: “decreasing by 33.6%” - how such % was calculated? Which period it is for the denominator.

Authors: We have used the 1960 to 1979 as reference period, and have now added the range to the statement.

R1C26: Line 78: $p > 0.05$. what if a p value = 0.1? is the result sensitive to the p value cutoff?

Authors: Following your suggestion we have tested the cutoff of $p = 0.1$, and found it had no influence on the results.

Figure R2. Temporal dynamics of the climate sensitivities of the atmospheric CO₂ growth rate (CGR) and net ecosystem exchange (NEE), similar to Fig. 1. A solid circle marker indicates significant ($p < 0.05$) sensitivities of CGR to climate variables in that 20-year window, while open circles indicate insignificant ($p > 0.05$) sensitivities.

R1C27: Line 78: replace “negligible” role – insignificant. Also “suggesting a negligible role of tropical water availability in controlling the interannual variability of CGR.” – contradicting with conclusions made later.

Authors: We have made the change as suggested.

We apologize for the confusion here. In this study, we examined the water influence on CGR variability at two temporal scales: the interannual time scale and the long-term (bi-decadal) time scale. While our best empirical model suggests that at the interannual time scale, water availability does not show significant influence on CGR IAV (Fig. 1 and Fig. S1), at the long-term time scale the water availability strongly influences STD_{CGR} . The apparent contradiction here reflects the different influences at two different time scales.

R1C28: Fig. 1a: add which statistical model is used for each curve in the caption.

Authors: Thank. We have added M1 for and M3 in the legend.

R1C29: Line89: replace “non-significant” to “insignificant”

Authors: This has been revised as suggested.

R1C30: Line 90: 1000 bootstrap estimates: per 20yr window? no info at all for how bootstrap simulations were conducted.

Authors: We followed the method introduced in Wang et al. 2013. In each 20-year time window, we bootstrapped CGR 100 times using its uncertainty, and calculated the climate sensitivities in each bootstrap. We have added a citation to the paper here to indicate our

method used.

R1C31: Line 91: uncertainty quantification: standard error vs standard deviation. CI.

Authors: we have changed throughout the manuscript to use standard error where possible. CI here is used to describe the confidence range for the linear regression, as the use of standard error or deviation is not applicable here.

R1C32: Line 93: first value? Value in the first year in each 20yr window?

Authors: We have revised the statement to say “ STD_{CGR} and STD_{NEE} were calculated for every 20-year window from 1959 to 2016, and normalized by their respective first values (i.e., the STD_{CGR} and STD_{NEE} of 1959 to 1978).”

R1C33: Line 99-100: “We found that both DGVMs and the machine learning products mischaracterized the 99 temperature sensitivity (γ_{NEET}) of terrestrial net ecosystem exchange (NEE)” – not surprising, but how about LULC? – see my comments above.

Authors: Please see our response to R1C3 and associated analysis, where we suggest that long-term changes in LULC did not contribute much to the long-term changes in STD_{CGR} .

R1C34: Line 104-108: Nice to see this clear statement! Fully agree.

Authors: Thank you.

R1C35: Line 115-116: “we found that the STD_{CGR} increased 34.8% from the 1960s to the 2000s, and then decrease slightly by 13.2% in the recent decade” – what is the denominator in calculating such % changes?

Authors: We apologize for the lack of clarity. We have added the absolute values of STD_{CGR} in the sentence to indicate the denominator is the STD_{CGR} value from the previous period.

“Indeed, we found that the STD_{CGR} increased 34.7% from 0.98 PgC yr⁻¹ in 1960-1979 to 1.32 PgC yr⁻¹ in 1985-2004, and then decreased slightly by 14.4% to 1.13 PgC yr⁻¹ in 1997-2016 (Fig. 1c)”.

R1C36: Line 116-117: such a change in STD_{CGR} explained 65% of the variance in and such a change in temperature sensitivity. – again, I don’t think STD_{CGR} is the cause.

Authors: We have revised it to “such a change in STD_{CGR} underlies the dynamics of γ_{CGR}^T we detected ($r^2 = 0.65$; Fig. 1d).” Please see our response to R1C10, where we justify why we use STD_{CGR} .

R1C37: Line 119-120 and 440: LULC changes from Global Carbon Budget, which may have a very smoothed CGR. See my comments above.

Authors: Please see our response to R1C3, where we also used the LULC emissions from Yue et al. 2020 in our analysis to support our argument.

R1C38: Line 123-124: “The long-term dynamics of STD_CGR over the past 60 years are independent of the year-to-year climate variability but driven by a long-term factor.” – the claim seems very odd, and even contradicting with previous statements.

Authors: We apologize for the confusion and have removed the statement.

R1C39: Line 125: “long-term water availability” - can be confusing. What is long-term?

Authors: “long-term” refers to 20 years in the study, here the water availability means the dynamic of the 20-yr climatological mean of precipitation, TWS or soil water content.

R1C40: Line 138: “long-term water availability” should be climatological mean?

Authors: Yes, that is right. We get the climatological mean of every 20-year to represent long-term water availability of that bi-decade.

R1C41: Line 141: uncertainty: 1 standard error again

Authors: We have revised the manuscript to consistently using standard error throughout.

R1C42: Line 142: area affected by droughts? What is the denominator? – global area?

Authors: We have revised the sentence to “the temporal dynamics of the percentage of tropical vegetated area affected by droughts of different intensities”. The denominator is the total tropical vegetated lands.

R1C43: Line 143-144: Percentile of precipitation – to establish the long-term statistics, using each 20yr moving window, or across the entire period?

Authors: We have revised the sentence to “drought intensity is defined by the percentile of local monthly precipitation across the whole study period (1959-2016)”.

R1C44: Line 147: why both linear regressions with and without intercept were examined?

Authors: We provided both linear regressions to facilitate the time-to-space substitution analysis in the next step. Having a negative intercept causes unrealistic extrapolations in the time-to-space substitution when drought-affected area is small.

R1C45: Line 149: The relationship in panel C is sensitive to the cutoff value used in the definition of extreme drought, as shown in panel d. But this was interpreted as mild drought has a limited influence on STD_CGR (line 178-179). Also, When drought intensity is low, is the

correlation of STD_CGR and area affected by extreme drought still significant? Why suddenly r not R2?

Authors: Fig. 2d demonstrates the correlations between drought affected area and STD_{CGR} . When we only study the extreme droughts (x axis drought intensity 5%, 10%), the correlation is strong (exemplified in Fig. 2c). However, as we add in the non-extreme droughts affected area, the correlations between droughts-area and STD_{CGR} became weaker. It suggests non-extreme droughts (i.e., mild droughts) are not as closely related to STD_{CGR} as extreme droughts.

The correlation between STD_{CGR} and area affected by extreme droughts is significant ($p < 0.05$; Fig. 2c), but the relationship with STD_{CGR} is insignificant ($p > 0.05$) for non-extreme droughts (Fig. 2d).

We used r instead of r^2 in the panel mainly help to demonstrate the difference between DGVMs and CGR results. While STD_{CGR} is positively correlated with drought-affected area, STD_{NEE} from DGVM is negatively correlated with drought-affected area, highlighting a striking difference between models and observation in their representation of extreme drought influences on CGR.

R1C46: Line 161: Meteorological drought (line 161) is short term, in contrast to the multiple year lagged effect or long-term mean TWS stated above

Authors: We used the definition of meteorological drought to identify droughts over time, but examine the changes in their frequency (i.e., duration weighted drought affected area) over moving 20-years windows when connecting droughts to STD_{CGR} . Please also see our response to R1C5 for more details.

R1C47: Line 164: drought-affected area should be biomass weighted.

Authors: Thank you. Please see our response to R1C4.

R1C48: Line 166: drought duration – very different from drought frequency defined in the title. No information provided in the method section

Authors: Please see our response to R1C2. We have revised our title to remove the use of drought frequency.

R1C49: Line 166-170: percentage change, what is the denominator, given the different time period?

Authors: Thanks for the comment. We meant the percentage of tropical vegetated area affected by extreme droughts; the denominator is the total area of vegetated area in the

tropics for all periods. We have revised the sentence to clarify that "...% of tropical vegetated land surface...".

R1C50: Line 170-171: "the long-term change in tropical water availability was primarily driven by the increasing occurrence of extreme droughts." – this does not sound logical, and the previous statements do not really directly support this claim.

Authors: We have removed the statement.

R1C51: Line 176-178: "outsize influence" – really depends on where the drought occurs, as later acknowledged by authors – "extreme droughts happened unevenly over time and space, and thus contributed differently to the variation of STD_{CGR} "

Authors: Here we are highlighting -- compared to the area under extreme drought influence (6%-9%), the variances in STD_{CGR} explained by the extreme drought area is outsized (75%). Please also see our response to R1C4 where we used a data-driven NEE dataset as a spatial weight to examine the regional contribution to STD_{CGR} .

R1C52: Line 188-189: when comparing tropical Africa and tropical America and Asia, the extreme impacted drought area was calculated using the total pan tropical area (which is consistent across three regions) or the respective total tropical area in each continent? Also, is the relationship found in Fig. 2-3 sensitive to the definition of tropical area, which was never clearly defined?

Authors: We apologize for not defining tropics early on. We have added in the abstract and the introduction that the study area is "tropical vegetated land surface (23°S – 23°N)".

The denominator used to calculate the extreme drought affected area for each continent was the total tropical vegetated land area throughout the manuscript. We have added "...% of tropical vegetated land surface" in L202 to indicate this.

R1C53: Line 189: tropical America Africa?

Authors: Sorry for the typo. We have corrected it to "tropical America".

R1C54: Line 199-202: this does not make sense, see my comments above

Authors: Please see our response to the major comment R1C4, where we used data-driven flux as weight when quantifying regional contribution to STD_{CGR} .

R1C55: Line 203: how semi-arid/arid ecosystems vs rainforests were defined? What datasets were used?

Authors: We used MODIS land cover product to delineate semi-arid/arid ecosystems and tropical forests. Please see our response to R1C7.

R1C56: Line 205-209: cannot really compare across continents IF using the scaling approach defined in line 199-202. Very different biomes. Even rainforests are highly different between Amazon and Congo.

Authors: Please see our response to R1C4, where we added a data-driven flux as the spatial weight to differentiate biomes.

R1C57: Line 218: The extreme-induced STD_{NEE} (Fig. 3b, d, f) were estimated using the relationship between drought-affected area and STD_{CGR} (Fig. 2c). STD_{NEE} and STD_{CGR} totally different things and were never validated. How to interpret the differences in Pan Tropic and CGR?

Authors: Thanks for the comment. We have noted in the introduction that previous studies have found that the variation in CGR is mostly driven by the variation in NEE in the terrestrial ecosystems (Cox et al., 2013) and most of the variation (70%~85% depending on methods) in terrestrial NEE is generated in the tropics (Piao et al., 2020). Therefore, the STD_{CGR} mainly reflects the dynamic of global and pan-tropical STD_{NEE} . In this study, we used global STD_{NEE} throughout the analysis, but see some results based on tropical STD_{NEE} in Fig. S5. Meanwhile, we noticed the contribution from ocean-atmosphere CO_2 exchange or other sources do not explain the STD_{CGR} (Fig. S2).

R1C58: Line 225-226: STD_{CGR} highly correlated with temperature, and also highly correlated with decadal scale mean TWS. But what is the cause, what is the consequence, and what is the constraint? Here everything is based on correlational analysis.

Authors: Sorry for the lack of clarity here. We did not suggest STD_{CGR} is correlated with temperature. STD_{CGR} (the change in the magnitude of CGR IAV) is related to bi-decadal mean water availability, CGR IAV is related to temperature.

R1C59: Line 226-227: we attribute the strong coupling between STD_{CGR} and long-term water availability to the effects of extreme drought on tropical terrestrial carbon cycling. But extreme drought is not equivalent to long-term water availability. See my comments above.

Authors: Please see our response to R1C5, where we outline how we did not use long-term water availability to define drought.

R1C60: Line 236: our results help reconcile conflicting reports on the water sensitivity of CGR – not sure if this goal is met.

Authors: We have removed the statement, as it distracted readers from our main goal – to explain the changes in STD_{CGR} .

R1C61: Line 241-242: “It is possible that extreme climate events – which are likely underrepresented by the relatively short and sparse eddy covariance measurements – led to structural and long-term changes that are undetected by flux sites” - not sure if this claim is really true. Most of the EC measurements so far are longer than 20yr window. I think the problem in FLUXCOM is more related to the upscaling approach, which cannot really conserve the temporal variability.

Authors: The median site record length in the FLUXNET database is 5 years (Fig. R3), and only one site (US-Ha1) has a 20 year record. In addition it is important to note that there is much less representativeness of EC sites in some key ecoregions (Jung et al., 2020). Though there are more and more long-term sites now, the number of EC sites with standardized gap-filled observations longer than 20 years is still quite limited (Hawkins et al., 2020), even more so when the FLUXCOM project was conducted (*ca.* 2013).

Figure R3. The distribution of the length of the observational record at each of the 206 sites in the FLUXNET 2015 open access database. The vertical red line indicates the median site record length (5 years).

R1C62: Line 255-256: our results show that extreme droughts also influence CGR at longer time scales by amplifying STD_{CGR} and temperature sensitivity. How so? This paper does not really examine this amplification?

Authors: We have shown that with more extreme droughts are associated with larger STD_{CGR} . Since the inferred long-term change in the apparent temperature sensitivity is dependent on the changes in STD_{CGR} and the magnitude of temperature variability (STD_{tair}), and STD_{tair} did not change as much (Fig. S2b), it logically follows that enhanced STD_{CGR} underlies the enhanced apparent temperature sensitivity.

R1C63: Line 290-291: Our analysis shows that long-term tropical water availability is negatively

related to the changes in STD_{CGR} , which explains the CGR pronounced variations in temperature sensitivity, as CGR is more variable when there are more extreme droughts. – this claim is self-contradicted, and also contradicted to Fig. 2a

Authors: Sorry for the confusion. We have removed the statement to focus on the positive correlation between STD_{CGR} and drought-affected area. We feel the misunderstanding is due to a typo in the original statement where we mistakenly describe a correlation between STD_{CGR} and long-term tropical water availability. We apologize for that.

R1C64: Line 318: monthly soil water content: uncertainty of SWC calculated from the SPLASH model?

Authors: Unfortunately, the SPLASH model does not provide uncertainty estimates of annual SWC.

R1C65: Line 359: satellite-based TWS: no further info. where is this from? GRACE is only available since 2002

Authors: Please see our response to R1C7, where we have added more description of the reconstructed GRACE TWS dataset.

Reviewer #2 (Remarks to the Author):

R2C1: Summary:

This is a review of the manuscript Tropical drought frequency influences long-term dynamics of the temperature sensitivity of the global carbon cycle (NCOMMS-21-08853) submitted to Nature Communications. This manuscript is well suited to the topics of Nature Communications as it is an investigation of why the sensitivity of atmospheric CO₂ growth rate (CGR) to tropical temperatures – already established in literature – changes over time. The manuscript uses observations of atmospheric CO₂ concentration along with observations/reanalysis of precipitation, temperature, radiation, and water availability to calculate the sensitivity of CGR to tropical temperatures using multiple model formulations for a detailed model selection. From this effort they show that the models are generally consistent in their retrieval of the sensitivity of CGR interannual variation with Temperature and water (precipitation or storage). They perform these regressions along a moving 20 year window to show that the sensitivity of CGR to Temperature has changed over time, while the (relatively low) sensitivity of CGR to Water has not. Additionally, these results were compared to similar results (using net ecosystem emissions rather than CO₂ concentrations) from dynamic vegetation models and FluxCom.

Authors: Thank you. We appreciate the accurate summary and the positive comments from the reviewer.

R2C2: They then correlated the change in the sensitivity of CGR to a change in CGR standard deviation along with a long term change in water storage and the extent of extreme drought. Their conclusion is that the change in land area impacted by extreme drought drove a large part of the increased sensitivity of CGR to temperature, particularly in Tropical Africa.

I found this to be a well written manuscript on an important topic relating to how the tropical biosphere function will respond to long term change. However, I have reviewed a previous version of this manuscript and continue to be concerned by reporting r-squared and significance statistics based on running regressions in the main manuscript. Each regression coefficient calculated in a running regression includes a large amount of non-independent data. Accounting for this will likely significantly – and I would suggest appropriately – reduce the statistical power of this analysis. Though this is a major concern, I do think that the inference in this manuscript may still be supported and will benefit from a more conservative analysis. These concerns relate to both the analysis presented in Figure 1d (relating standard deviation of CGR and sensitivity of CGR to Temperature) and Figure 2a and 2c (relating area of extreme drought to change in sensitivity of CGR to Temperature).

Authors: We very much appreciate your continued efforts to help improve our manuscript. As the reviewer might remember we had addressed the autocorrelation issue in a previous

revision, but that aspect of the analysis was removed in a later revised version. Following the suggestion, we have now reexamined the autocorrelations of time series in our analyses and updated relevant figures and statements.

R2C3: I suggest a major revision for the calculation of the change of sensitivity over time to account for the autocorrelation inherent in a running calculation or do away with it entirely in supporting the inference of the manuscript. Other minor revisions detailed below.

Authors: Thank you for the suggestion. We have made the following revisions to account for autocorrelation effect in our correlation analysis.

1. In the methods, we added a new section to describe the removal of autocorrelation (L452-475):

“Remove the influence of autocorrelation. Autocorrelation is the correlation of a time-series with a delayed copy of itself. Since we used a moving window to calculate STD_{CGR} , the value of STD_{CGR} in a window is not independent from its counterparts in adjacent windows, leading to autocorrelation in the time series of STD_{CGR} . In addition, since we calculated long-term variability in tropical water availability (e.g., \overline{MAP} , \overline{SWC}) using the same moving window, it can also generate some autocorrelation in these time series. We have taken the following steps to address the issue of autocorrelation:

1) First, we introduced the Durbin-Watson indicator (DW) to evaluate the autocorrelation of these times series. We estimated the lag 1 autocorrelation in the residuals (e_i) from the ordinary linear regression of the variables of interest (e.g., STD_{CGR} , \overline{MAP}) to time, where the coefficient in the regression is ρ , as in $e_i = \rho e_{i-1} + r_i$. The DW tests the null hypothesis that residuals are uncorrelated ($\rho = 0$), against the alternative hypothesis that autocorrelation exists ($\rho \neq 0$) (Equation 1).

$$DW = \frac{\sum_{i=2}^n (e_i - e_{i-1})^2}{\sum_{i=1}^n e_i^2} \quad (1)$$

Where n is the number of observations, e_i is the i th residual of the linear regression of a target variable to time. The Durbin-Watson indicator (DW) is a value ranging between 0 and 4, where 0 means positive autocorrelation, 2 means no autocorrelation and 4 means negative autocorrelation.

2) We applied the Cochrane-Ocruitt procedure to adjust the variables of interests (y_i ; e.g., STD_{CGR} , \overline{MAP} , \overline{SWC} , \overline{TWS} , \overline{MAT} , \overline{VPD}) to STD_{CGR_adj} , \overline{MAP}_{adj} , \overline{SWC}_{adj} , \overline{TWS}_{adj} , \overline{MAT}_{adj} and \overline{VPD}_{adj} where $y_{i_adj} = y_i - \rho y_{i-1}$. After the procedure, we found the autocorrelations in STD_{CGR_adj} , \overline{MAP}_{adj} , \overline{SWC}_{adj} , \overline{TWS}_{adj} , \overline{MAT}_{adj} and \overline{VPD}_{adj} were largely removed since their DW values were

close to 2 -- 1.6, 1.3 ± 0.1 , 1.2, 1.3, 1.9 and 1.9 ± 0.03 , respectively. Among them, \overline{MAP}_{adj} and \overline{TWS}_{adj} were negatively correlated to STD_{CGR_adj} with significance level of $p < 0.01$ and $p = 0.08$ (Fig. 2a), respectively, meaning the negative impact of long-term water availability on STD_{CGR} we found is significant regardless of autocorrelation.”

2. We have update Fig. 2a based on the analysis above, where we used the Cochrane-Ocrutt procedure to remove the autocorrelations in time series (hollow bar) and relevant statement (L371-372). We found after that autocorrelation correction the relationship between (\overline{MAP}) and STD_{CGR} remains significant ($p < 0.01$), supporting our next step of investigation on droughts.

Figure 2a. The variance in STD_{CGR} explained by long-term water availability or temperature in the tropics, as represented by 20-year average terrestrial water storage (\overline{TWS}), soil water content (\overline{SWC}), mean annual precipitation (\overline{MAP}), vapor pressure deficit (\overline{VPD}) and mean annual temperature (\overline{MAT}). The error bars indicate the uncertainty (one standard error) in r^2 when using alternative precipitation and temperature datasets; the hollow bars indicate the variance in STD_{CGR} explained by long-term tropical water availability or temperature, after accounting for autocorrelations in STD_{CGR} and all climate variables using the Cochrane-Ocrutt procedure (see Methods). ‘*’ means $p < 0.1$, ‘**’ means $p < 0.01$.

3. We have added a new supplementary Fig. S3 to address the autocorrelations of the time series (STD_{CGR} , γ_{CGR}^T and γ_{CGR}^W and extreme drought-affected area) and the relationships between them (as a supplementary to Fig. 2c). We applied the similar Cochrane-Ocrutt procedure and found after the autocorrelation adjustments, the relationship between STD_{CGR} and area-affected area (Fig. S3a) and the relationship between extreme drought-affected area and STD_{CGR} (Fig. S3b) remain significant ($p < 0.01$).

Figure S3. Remove the autocorrelations in Fig. 1d and Fig. 2c. We used the Cochran-Ocrutt procedure (see Methods) to remove autocorrelations in time series. (a) The relationships between adjusted climate sensitivities of CGR (i.e., $\text{adj. } Y_{\text{CGR}}^{\text{T}}$ and $\text{adj. } Y_{\text{CGR}}^{\text{W}}$) and adjusted STD_{CGR} (shading: 95% confidence interval); (b) The relationship between adjusted STD_{CGR} and adjusted extreme drought-affected area (shading: 95% confidence interval).

Reviews of [figures] and [lines] below.

Minor revisions:

R2C4: [189] Possible typo, correct for clarity: ‘tropical America Africa’

Authors: We apologize for the typo, and have corrected it to “tropical America”.

R2C5: [241 - 244] When discussing FluxCom and the lack of sensitivity, it should be noted that FluxCom interannual variability is also reduced compared to the flux towers that underpin the product (Tramontana et al. 2016). Additionally, there are very few flux towers available or included in FluxCom in the tropics.

Authors: Thank you for the suggestion, we have added the following statements in the Discussion to acknowledge the issues (L307-321).

“FLUXCOM – a data-driven machine learning NEE product - was also unable to fully capture the long-term dynamics in $Y_{\text{CGR}}^{\text{T}}$ and STD_{CGR} (Fig. 1a,b). The underrepresentation of extreme droughts in the relatively short eddy covariance measurements (i.e., limited sites have more than 10 years of records) (Hawkins et al., 2020) and the lack of sites in the tropics (Jung et al., 2020) could lead to structural and long-term changes being undetected, and consequently to the underestimation of tropical NEE variability and CGR climate sensitivities. Other than the impact of extreme droughts, we acknowledge that muted interannual variability of FLUXCOM product is also caused by the incapability of machine learning algorithms to capture low

frequency variations at the interannual time scale and the use of average remote sensing forcing(Jung et al., 2020). However, unlike DGVMs, FLUXCOM identified a weak yet positive relationship between extreme drought-affected area and STD_{NEE} (Fig. 2d), showing that it has a better representation of extreme drought effects than DGVMs. It is worth noting that NEE of tropical extreme drought affected regions from FLUXCOM can show a sensitivity of STD_{NEE} to drought that is close to that of STD_{CGR} to drought (Fig. S5). To further improve the prediction of machine learning algorithms of CGR variability, one potential path is to use algorithms such as Long Short-Term Memory (LSTM) to incorporate the lagged effects of climate extremes into the simulation of terrestrial carbon fluxes(Besnard et al., 2019).”

R2C6: [257 – 261] This discussion of mechanisms would benefit from including discussion of vapor pressure deficits direct impact on gross primary productivity and its non-linear response to temperature. Increased Temperatures in the tropics should lead to increased variability of VPD even if the variability of Temperature stays the same.

Authors: Thank you for this suggestion. We agree with the reviewer that non-linearities of climate sensitivities should be acknowledged. We have added a new section in the Discussion (L269-280).

“In this study, we use nine competing linear models to derive γ_{CGR}^T and γ_{CGR}^W (Fig.S1). The values of γ_{CGR}^T and γ_{CGR}^W and their long-term dynamics are highly influenced by the types of linear models and climate data used. Statistically, our result shows that γ_{CGR}^W is insignificant as long as the models include tropical temperature as a predictor. However, when using univariate linear models we note γ_{CGR}^W is significant (Fig. S1h), caused by the strong correlation between ΔMAT and ΔTWS (Piao et al., 2020) or tropical lagged precipitation(Wang et al., 2016). Previous studies have reported non-linear responses of the tropical terrestrial carbon fluxes to temperature(Tan et al., 2017), VPD(Green et al., 2020), precipitation(Guan et al., 2015) and soil moisture(Green et al., 2019), which question the common practices of using linear models to derive γ_{CGR}^T and γ_{CGR}^W (Humphrey et al., 2018; Wang et al., 2013, 2014), though non-linear models may not be statistically stronger than linear models with less degrees of freedom for fitting. To avoid the uncertainties in climate sensitivities incurred by the type of models and data used, we use STD_{CGR} , which is calculated from perhaps the least uncertain term in the global carbon cycle (i.e., CGR), as a proxy for γ_{CGR}^T to examine the long-term dynamics.”

Since either VPD or MAT can explain the STD_{CGR} (Fig. 2a), we did not specifically delve into the non-linear relationship between the two. We did notice that VPD is more variable than T in the tropics (Fig. S2b; the figure is available in our response to R2C7).

R2C7: [Figure 1d] The relationship between the standard deviation of CGR with the

Temperature simply due to the relative correlation of Temperature of with CGR – such that any increase in standard deviation will result in an increase in sensitivity (being the combination of correlation and the ratio of standard deviations of Temperature and CGR). Does the correlation between Temperature and CGR or the standard deviation of Temperature change at all through the time period?

Authors: Thanks for the comment, we have updated Fig. S2 to address this question.

The standard deviation of temperature (STD_{tair}) decreased by about 40% in the 1990s and then became steady afterwards. The decrease in STD_{tair} can increase γ_{CGR}^T if there is no change in STD_{CGR} . However, the correlation between STD_{tair} and γ_{CGR}^T ($r^2=0.52$) is weaker than that between STD_{CGR} and γ_{CGR}^T ($r^2=0.65$), showing that more changes in γ_{CGR}^T are driven by STD_{CGR} .

In addition, using a different source of temperature may result into a different STD_{tair} , which means more uncertainties in γ_{CGR}^T . Therefore, we focused our study objective on STD_{CGR} , which is a much better constrained term in the global carbon cycle.

The r^2 between temperature and CGR ranges from 0.17 to 0.65 in each 20-year windows, and they are all statistically significant ($p<0.01$) except the first segment (1959-1978), shown in Fig. S1g.

Figure S2. Temporal changes in the magnitude of the interannual variability of CGR, ocean uptake, land use change emissions, fire emissions, leaf area index (LAI) and climate variables. Interannual variability is indicated by one standard deviation of annual values (STD) within 20-year moving window. (a) CGR, ocean carbon uptake and land use change emissions are provided by the Global Carbon Budget. CY-LULC indicates the land use change emissions estimated by ORCHIDEE, which have considered the changes in carbon density and climate variability for the land use change emissions (Yue et al., 2020). Fire emissions are obtained from Global Fire Emissions Database (GFEDv4.1). Since fire emissions record only range from 1997 to 2020, we used 10-year moving window to calculate the STD of fire fluxes. The shadings of ocean uptake indicate one standard error of eight models. LAI is the GMMIS AVHRR3g LAI over the tropics from 1982 to 2016. (b) STD of air temperature (Tair), precipitation

(Precip), vapor pressure deficit (VPD) and soil water content (SWC) are normalised by the STD of first 20-year window (1959-1979) to make them comparable.

References

Tramontana, G. et al. Predicting carbon dioxide and energy fluxes across global FLUXNET sites with regression algorithms. *Biogeosciences* 13, 4291–4313 (2016).

Reviewer #3 (Remarks to the Author):

R3C1: This paper links variability in the rate of growth of atmospheric CO₂ to variability in tropical land surface temperature and assorted metrics of tropical land surface water availability. The authors find that variability in atmosphere CO₂ growth rate (CGR) increased over much of latter 20th century, but has recently decreased. The sensitivity of CGR to tropic temperature likewise increased during the late 20th century, but decreased during the early 21st century. The authors report that the sensitivity of CGR to tropical water availability is weak, but that the area of tropical land affected by extreme drought is strongly correlated with variability in CCR.

Authors: Thank you very much for reviewing our paper and sharing your insights on the topic.

R3C2: Overall, I found the methods appeared to be state of the art, and in general I was able to understand clearly what the authors did (but see a few notes on presentation below). I think the paper is interesting.

Authors: We appreciate the constructive comments from the reviewer and have used them to improve our manuscript.

I offer the following generally notes and reflections:

R3C3: First, I think the study would benefit from clearer attention to context, especially in terms of what's been done already on the topic. This is not the first study seeking to link variability in CGR or land carbon uptake to temperature in the tropics; the authors mention that their study is preceded by earlier work, but it would be nice if they could offer a summary of what has already been learned from the work focused on tropical climate, and also the complimentary literature suggesting it is actually dryland C uptake sensitivity that drives interannual variability in the global carbon sink anomalies (e.g. Poulter et al. 2014). Then, the authors could identify the specific knowledge gaps that remain (i.e. identify the questions that will be answered in this study), and also discuss who benefits from the knowledge that will be gained? It seems the primary audience is the modeling community...are there other groups of scientists or stakeholders who should be interested specifically in the links between variability in CGR (as opposed to the mean trend) and tropical climate?

Authors: Thanks for the suggestions. We have carefully used the comments from the reviewer to clarify our statements in the introduction.

1. Recent evidence has shown that the drylands-dominant view and tropic-dominant view on interannual variability (IAV) of CO₂ growth rate (CGR) are not at odds with each other, as extra-tropical drylands explained only <2% of the net land carbon flux IAV (Piao et al., 2020). Most of

global drylands impacts on CGR IAV were stemmed from the tropical drylands, at least according to DGVMs and atmospheric inversions. In the revised manuscript we summarize the previous studies as follows (L36-40):

“...the majority of that variability is driven by changes in tropical climate (Keeling et al., 1995; Rödenbeck et al., 2018; Wang et al., 2013, 2014) through the influence of climate on tropical terrestrial ecosystems (Bousquet et al., 2000; Fan et al., 2019; Peylin et al., 2013), among which tropical forests (Cox et al., 2013) or semi-arid ecosystems (Ahlström et al., 2015; Poulter et al., 2014) (i.e., most of the variable semi-arid ecosystems are in the tropics (Piao et al., 2020)) or both (Piao et al., 2020) were reported to play the primary role”.

2. Though previous studies have intensively investigated the correlations between CGR and tropical temperature and water, and derived the temperature and water sensitivity of CGR ($\gamma_{\text{CGR}}^{\text{T}}$ and $\gamma_{\text{CGR}}^{\text{W}}$), it remains unexplained why $\gamma_{\text{CGR}}^{\text{T}}$ would change over the long-term. The niche of our study is to examine long-term changes in $\gamma_{\text{CGR}}^{\text{T}}$ (thus STD_{CGR}), rather than discussing the relative importance of temperature or water in determining the CGR interannual variability.

We have revised the title and the Introduction to better reflect our objective. We updated the title to “Tropical extreme droughts associated with long-term increase in global carbon cycle variability”.

3. Motivated by the comments, we have extensively revised the discussion to explore why DGVMs and FLUXCOM are unable to reproduce the long-term dynamics of CGR variability (i.e., STD_{CGR}). Please see details in the response to R3C11 or the update discussion in L293-323.

R3C4: There seem to be a substantial number of scientists in the field who are seeking for “silver bullet” explanations for variability in the land carbon sink, and that this has been going on quite some time. But the problem is that results from one study to the next are often contradictory, except in their conclusion that the “models are wrong.” So basically, I’m looking for a clear explanation of why we need another study that comes to the same conclusion right now.

Authors: Please see our second point in the response to your comment (R3C3) above, where we identify the knowledge gap and highlight that our goal is to explain the changes in $\gamma_{\text{CGR}}^{\text{T}}$ (and thus STD_{CGR}) at bi-decadal time scale, not the CGR variability at the interannual time scale.

R3C5: Next, I have two observations about methodology and interpretation. First, when specifically considering the relationship between “extreme drought” and CGR variability, what

the authors have shown is correlation, but not causation. Is it possible that the occurrence of extreme drought in the tropics is itself driven by other mechanisms (e.g. El Niño or other global scale phenomena) which might affect the land carbon sink in many ways, not just by making drought more likely in the tropics? The strong correlation between the area of tropical land under extreme drought and the $\text{std}(\text{GCR})$ (Fig 2c) is certainly striking. But it is really possible that this small area of land could have such an outsized influence on the global land carbon sink? It seems that, at a minimum, some back-of-the-envelope calculations based on our best guesses of NEE in the tropics, and its sensitivity to drought stress, would be informative. For example, consider a hypothetical case where one year, none of the tropical land area is under drought stress, and another where let's say 8% of the land area is under drought stress. What is the expected change in the tropical NEE, and what fraction of global NEE does that change represent? Are the results coherent with the slope of the line appearing in Fig 2c?

Authors: Thank you for the reflections, which have motivated us to conduct the following analysis:

1. We agree with the reviewer that the tropical drought frequency was potentially related to ENSO phases, and have added the following section to address it (L284-291).

“Tropical extreme droughts developed preferentially during El Niño events (Jiménez-Muñoz et al., 2016; Lyon, 2004). Therefore, the drought- STD_{CGR} correlation can also be interpreted as an El Niño Southern Oscillation (ENSO)- STD_{CGR} correlation. We use the Multivariate ENSO Index (MEI) to represent the frequency and strength of El Niño, and find MEI is positively related with ΔCGR , STD_{CGR} and the extreme drought-affected area ($p < 0.01$; Fig. S4). The test shows ENSO not only modulates CGR at the interannual time scale (Keeling et al., 1995), but also enhances the magnitude of CGR variability in periods with more frequent and stronger El Niños by increasing extreme drought frequency. It is worth noting that MEI ($r^2 = 0.68$, Fig. S4b) does not explain as much long-term variability in CGR as the extreme drought area ($r^2 = 0.75$, Fig. 2c)”

Figure S4. ENSO effects on annual CGR anomalies (ΔCGR), STD_{CGR} and extreme drought-affected area. Multivariate ENSO Index (MEI; <https://psl.noaa.gov/enso/mei.old/table.html>) is used to indicate ENSO phases, where positive values mean warm, El Niño events. Shadings indicate 95% confidence intervals.

2. We agree with the reviewer that it is helpful to examine modelled NEE (i.e., those from DGVMs and FLUXCOM) and see if extreme droughts can induce outsized NEE changes in models, and whether the sensitivity of STD_{NEE} to drought is similar to what we see from the atmospheric CO_2 signal.

In fact, we had done so in Fig. 2d, where we demonstrated that DGVMs and FLUXCOM modelled STD_{NEE} are negatively or weakly related to area affected by different types of tropical droughts. We have now added a new figure to expand the analysis.

We obtained NEE of different regions (i.e., global, tropical and tropical drought affected regions) and derived their respective STD_{NEE} , and then checked the drought- STD_{NEE} relationships in comparison with the drought- STD_{CGR} relationship (Fig. S5).

Figure S5. The correlations between STD_{CGR} (STD_{NEE}) and extreme drought-affected area. STD_{NEE} were calculated based on either global annual NEE, tropical annual NEE or tropical drought affected regions' annual NEE. Annual NEE were obtained from DGVMs and FLUXCOM. We used the ensemble mean NEE from each group of models. Shadings indicate 95% confidence intervals.

DGVMs drought- STD_{NEE} correlation at the global and the tropical scales are negative or insignificantly positive (Fig. S5b). Though there is a significant positive drought- STD_{NEE} for tropical drought affected regions, the sensitivity of STD_{NEE} to drought is too small to explain the outside effect of droughts on STD_{CGR} (0.01 vs 0.19, the slopes of panel a and panel b red line).

FLUXCOM performs better than DGVMs in capturing the positive drought- STD_{CGR} (STD_{NEE}) correlations (Fig. S5c). In particular, we found the sensitivity of STD_{NEE} to drought is close to the sensitivity of STD_{CGR} to drought when we only looked at NEE from extreme drought affected regions (0.13 vs 0.19, the slopes of panel a and panel c red line). FLUXCOM also shows that a large portion of global STD_{NEE} is from tropical drought regions STD_{NEE} . We have added the figure and above statement in the Discussion (L314-318).

R3C6: Speaking of linear slopes, that brings me to my second comment about methodology. I felt that the determination of the sensitivities of CGR to temperature and water availability using multiple linear regressions (e.g. Table 1) was overly empirical. Is a simple linear regression really the best way to approximate processes that are known to be highly non-linear? Consider, for example, the case of water storage, which we know will have a threshold-type relationship with NEE...when soil is well-watered, we would expect a priori that NEE should be insensitive to variability in soil water; on the other hand, when its dry, the relationship between the two will become more pronounced. Should we really be drawing a line through data that represent a

threshold-driven process? Likewise, with respect to temperature, again...is a linear relationship expected? Or perhaps its more exponential, reflecting the exponential relationship between respiration and temperature, compounded by reductions in photosynthesis when T is high. In any event, I admit its hard for me to understand why results of a simple linear regression should be interpretative as informative of the apparent “failure” of more complex and mechanistic earth system models.

Authors: We agree that non-linear climate sensitivities are possible, and have added the following text to the discussion section to explore potential implications (L271-282):

“In this study, we use nine competing linear models to derive $\gamma_{\text{CGR}}^{\text{T}}$ and $\gamma_{\text{CGR}}^{\text{W}}$ (Fig.S1). The values of $\gamma_{\text{CGR}}^{\text{T}}$ and $\gamma_{\text{CGR}}^{\text{W}}$ and their long-term dynamics are highly influenced by the types of linear models and climate data used. Statistically, our result shows that $\gamma_{\text{CGR}}^{\text{W}}$ is insignificant as long as the models include tropical temperature as a predictor. However, when using univariate linear models we note $\gamma_{\text{CGR}}^{\text{W}}$ is significant (Fig. S1h), caused by the strong correlation between ΔMAT and ΔTWS (Piao et al., 2020) or tropical lagged precipitation (Wang et al., 2016). Previous studies have reported non-linear responses of the tropical terrestrial carbon fluxes to temperature (Tan et al., 2017), VPD (Green et al., 2020), precipitation (Guan et al., 2015) and soil moisture (Green et al., 2019), which question the common practices of using linear models to derive $\gamma_{\text{CGR}}^{\text{T}}$ and $\gamma_{\text{CGR}}^{\text{W}}$ (Humphrey et al., 2018; Wang et al., 2013, 2014), though non-linear models may not be statistically stronger than linear models with less degrees of freedom for fitting. To avoid the uncertainties in climate sensitivities incurred by the type of models and data used, we use STD_{CGR} , which is calculated from perhaps the least uncertain term in the global carbon cycle (i.e., CGR), as a proxy for $\gamma_{\text{CGR}}^{\text{T}}$ to examine the long-term dynamics.”

R3C7: Finally, I was surprised to see so much emphasis on drought legacy effects in the discussion, as this wasn't really a focus of the analysis. On lines 254-257, the authors state “Several studies have suggested that a few extreme events explained a significant amount of the variance in land-atmosphere carbon exchange, at seasonal or interannual time scales. However, our results show that extreme droughts also influence CGR at longer time scales by amplifying STD_{CGR} .” Maybe I'm missing something, but I don't see how a relationship between extreme droughts and STD_{CGR} necessarily implies an important role for legacy effects. This area of the discussion would benefit from a review of independent evidence, even from site level studies, of the extent to which drought legacies characterize the dynamics of carbon exchange in the tropics. While I am not an expert on this topic, I would think that the biodiversity and mesic conditions that characterize much of the region should confer a fair amount of resilience to the impacts of extreme droughts.

Authors: We agree with the reviewer that more evidence is needed to delineate the role of drought-induced lagged effect on STD_{CGR} . Following the comment, we have removed the relevant statements in the Introduction and Results, and only kept a short discussion on lagged effect in the Discussion, stating that the lagged effect is potentially one of the reasons that DGVMs have trouble in capturing the long-term variability in CGR.

We appreciate the reviewer's thoughts on the resilience to drought due to high biodiversity. It provides one potential explanation for the changed drought frequency over time. With decreasing biodiversity in the tropics (Alroy, 2017; Giam, 2017), ecosystems may become less resilient to droughts and become more variable in their NEE (i.e., larger STD_{CGR}). Unfortunately, we do not have a good historical record of large-scale biodiversity in the tropics to test this hypothesis.

A few more minor comments:

R3C8: Abstract: Would it be possible to write an abstract that does not rely so strongly on symbols and abbreviations? It took me a long time to get through it, as I had to first learn what the different symbols referred to. A plain language description would seem more appropriate.

Authors: Thank you for the comment. We agree and have update the abstract to reduce the number of acronyms.

R3C9: As I'm sure the authors know, a study was published in Nature (Humphrey et al. 2021) recently that seems very relevant to the present study...in particular, the paragraph of Humphrey et al that contains the sentence: "Thus, although the IAV in global land carbon uptake has been empirically found to be sensitive to tropical mean temperature in numerous studies, our results suggest that this sensitivity does not represent a strong mechanistic link..." The authors of this study will probably need to respond accordingly.

Authors: Thank you for directing us to the new paper VH2021 (Humphrey et al., 2021). We have added the following statement to acknowledge it.

"Recent evidence demonstrated that γ_{CGR}^T and γ_{CGR}^W are related due to the land-atmosphere coupling by soil moisture(Humphrey et al., 2021), suggesting a potential change of γ_{CGR}^W over time."

In this round of revision, we have tightened up our study to focus only on explaining the long-term dynamics of STD_{CGR} . We have steered away from discussing the relative importance of temperature/water on CGR as many have looked into that, including VH2021.

Below we showed that even using CMIP5 models that consider land-atmosphere coupling -- similar to those used in VH2021-- models still cannot capture the long-term dynamics of STD_{CGR} .

Figure R2. The changes in the magnitude of atmospheric CO_2 growth rate variability and net ecosystem exchange variability (STD_{CGR} and STD_{NEE}). NEE were estimated by DGVMs, CMIP5 (historical) and FLUXCOM. STD_{CGR} and STD_{NEE} were calculated for every 20-year window from 1959 to 2016, and normalized by their respective first value.

R3C10: The decrease in CGR variability over the past two decades is really interesting, and seems like a good source of novelty for this study when compared to similar studies conducted before this declining trend became so apparent. The authors might consider emphasizing this aspect of their results more strongly. That the shift in slope occurs is mentioned in the abstract, but the reasons for it are not discussed.

Authors: Thank you for this suggestion. We found that the decrease in STD_{CGR} over the past two decades was driven by decreased area of extremes droughts in tropical Africa and Asia, and in semi-arid ecosystems. We have highlighted the discovery in the Abstract (L27-28) and relevant places in the Results.

“The historical increase in STD_{CGR} were driven by more frequent droughts in all continents and biomes but the recent decrease in STD_{CGR} were mostly due to decreased drought-affected areas in tropical Africa and Asia, and semi-arid ecosystems.”

R3C11: The authors spend a fair amount of ink describing how their results highlight model deficiencies. But what about the FLUXCOM product? Any recommendations for how that might be improved?

Authors: Thank you for the suggestion, we have rewritten a section in the Discussion on the FLUXCOM product and how to improve it (L309-323).

“FLUXCOM – a data-driven machine learning NEE product - was also unable to fully capture the long-term dynamics in $\gamma_{\text{CGR}}^{\text{T}}$ and STD_{CGR} (Fig. 1a,b). The underrepresentation of extreme droughts in the relatively short eddy covariance measurements (i.e., limited sites have more than 10 years of records) (Hawkins et al., 2020) and the lack of sites in the tropics (Jung et al., 2020) could lead to structural and long-term changes being undetected, and consequently to the underestimation of tropical NEE variability and CGR climate sensitivities. Other than the impact of extreme droughts, we acknowledge that muted interannual variability of FLUXCOM product is also caused by the incapability of machine learning algorithms to capture low frequency variations at the interannual time scale and the use of average remote sensing forcing (Jung et al., 2020). However, unlike DGVMs, FLUXCOM identified a weak yet positive relationship between extreme drought-affected area and STD_{NEE} (Fig. 2d), showing that it has a better representation of extreme drought effects than DGVMs. It is worth noting that NEE of tropical extreme drought affected regions from FLUXCOM can show a sensitivity of STD_{NEE} to drought that is close to that of STD_{CGR} to drought (Fig. S5). To further improve the prediction of machine learning algorithms of CGR variability, one potential path is to use algorithms such as Long Short-Term Memory (LSTM) to incorporate the lagged effects of climate extremes into the simulation of terrestrial carbon fluxes (Besnard et al., 2019).”

References:

Humphrey, V., Berg, A., Ciais, P., Gentine, P., Jung, M., Reichstein, M., Seneviratne, S.I. and Frankenberg, C., 2021. Soil moisture–atmosphere feedback dominates land carbon uptake variability. *Nature*, 592(7852), pp.65-69.

Poulter, B., Frank, D., Ciais, P., Myneni, R.B., Andela, N., Bi, J., Broquet, G., Canadell, J.G., Chevallier, F., Liu, Y.Y. and Running, S.W., 2014. Contribution of semi-arid ecosystems to interannual variability of the global carbon cycle. *Nature*, 509(7502), pp.600-603.

Ahlström, A., Raupach, M.R., Schurgers, G., Smith, B., Arneeth, A., Jung, M., Reichstein, M., Canadell, J.G., Friedlingstein, P., Jain, A.K., Kato, E., Poulter, B., Sitch, S., Stocker, B.D., Viovy, N., Wang, Y.P., Wiltshire, A., Zaehle, S., Zeng, N., 2015. The dominant role of semi-arid ecosystems in the trend and variability of the land CO_2 sink. *Science* (80-). 348, 895 LP – 899.

Alroy, J., 2017. Effects of habitat disturbance on tropical forest biodiversity. *Proc. Natl. Acad. Sci. U. S. A.* 114, 6056–6061. <https://doi.org/10.1073/pnas.1611855114>

Besnard, S., Carvalhais, N., Altaf Arain, M., Black, A., Brede, B., Buchmann, N., Chen, J., Clevers,

- J.G.P.W., Dutrieux, L.P., Gans, F., Herold, M., Jung, M., Kosugi, Y., Knohl, A., Law, B.E., Paul-Limoges, E., Lohila, A., Merbold, L., Roupsard, O., Valentini, R., Wolf, S., Zhang, X., Reichstein, M., 2019. Memory effects of climate and vegetation affecting net ecosystem CO₂ fluxes in global forests. *PLoS One* 14, 1–22.
<https://doi.org/10.1371/journal.pone.0211510>
- Bousquet, P., Peylin, P., Ciais, P., Le Quéré, C., Friedlingstein, P., Tans, P.P., 2000. Regional changes of CO₂ fluxes of land and oceans since 1980. *Science* (80-.). 290, 1253–1262.
- Cox, P.M., Pearson, D., Booth, B.B., Friedlingstein, P., Huntingford, C., Jones, C.D., Luke, C.M., 2013. Sensitivity of tropical carbon to climate change constrained by carbon dioxide variability. *Nature* 494, 341–344. <https://doi.org/10.1038/nature11882>
- Fan, L., Wigneron, J.-P., Ciais, P., Chave, J., Brandt, M., Fensholt, R., Saatchi, S.S., Bastos, A., Al-Yaari, A., Hufkens, K., Qin, Y., Xiao, X., Chen, C., Myneni, R.B., Fernandez-Moran, R., Mialon, A., Rodriguez-Fernandez, N.J., Kerr, Y., Tian, F., Peñuelas, J., 2019. Satellite-observed pantropical carbon dynamics. *Nat. Plants*. <https://doi.org/10.1038/s41477-019-0478-9>
- Friedl, M.A., Sulla-Menashe, D., Tan, B., Schneider, A., Ramankutty, N., Sibley, A., Huang, X., 2010. MODIS Collection 5 global land cover: Algorithm refinements and characterization of new datasets. *Remote Sens. Environ.* 114, 168–182.
<https://doi.org/10.1016/j.rse.2009.08.016>
- Giam, X., 2017. Global biodiversity loss from tropical deforestation. *Proc. Natl. Acad. Sci. U. S. A.* 114, 5775–5777. <https://doi.org/10.1073/pnas.1706264114>
- Green, J.K., Berry, J., Ciais, P., Zhang, Y., Gentine, P., 2020. Amazon rainforest photosynthesis increases in response to atmospheric dryness. *Sci. Adv.* 6, 1–10.
<https://doi.org/10.1126/sciadv.abb7232>
- Green, J.K., Seneviratne, S.I., Berg, A.M., Findell, K.L., Hagemann, S., Lawrence, D.M., Gentine, P., 2019. Large influence of soil moisture on long-term terrestrial carbon uptake. *Nature* 565, 476–479. <https://doi.org/10.1038/s41586-018-0848-x>
- Guan, K., Pan, M., Li, H., Wolf, A., Wu, J., Medvigy, D., Caylor, K.K., Sheffield, J., Wood, E.F., Malhi, Y., Liang, M., Kimball, J.S., Saleska, S.R., Berry, J., Joiner, J., Lyapustin, A.I., 2015. Photosynthetic seasonality of global tropical forests constrained by hydroclimate. *Nat. Geosci.* 8, 284–289. <https://doi.org/10.1038/ngeo2382>
- Hawkins, L., Kumar, J., Luo, X., Sihi, D., Zhou, S., 2020. Measuring, Monitoring, and Modeling Ecosystem Cycling. *Eos (Washington, DC)*. 101. <https://doi.org/10.1029/2020EO147717>
- Humphrey, V., Berg, A., Ciais, P., Gentine, P., Jung, M., Reichstein, M., Seneviratne, S.I., Frankenberg, C., 2021. Soil moisture – atmosphere feedback dominates land carbon uptake variability. *Nature* 592. <https://doi.org/10.1038/s41586-021-03325-5>
- Humphrey, V., Zscheischler, J., Ciais, P., Gudmundsson, L., Sitch, S., Seneviratne, S.I., 2018. Sensitivity of atmospheric CO₂ growth rate to observed changes in terrestrial water storage. *Nature* 560, 628–631. <https://doi.org/10.1038/s41586-018-0424-4>
- Jiménez-Muñoz, J.C., Mattar, C., Barichivich, J., Santamaría-Artigas, A., Takahashi, K., Malhi, Y., Sobrino, J.A., Schrier, G. Van Der, 2016. Record-breaking warming and extreme drought in the Amazon rainforest during the course of El Niño 2015-2016. *Sci. Rep.* 6, 1–7.
<https://doi.org/10.1038/srep33130>
- Jung, M., Schwalm, C., Migliavacca, M., Walther, S., Camps-Valls, G., Koirala, S., Anthoni, P.,

- Besnard, S., Bodesheim, P., Carvalhais, N., Chevallier, F., Gans, F., S Goll, D., Haverd, V., Köhler, P., Ichii, K., K Jain, A., Liu, J., Lombardozi, D., E M S Nabel, J., A Nelson, J., O'Sullivan, M., Pallandt, M., Papale, D., Peters, W., Pongratz, J., Rödenbeck, C., Sitch, S., Tramontana, G., Walker, A., Weber, U., Reichstein, M., 2020. Scaling carbon fluxes from eddy covariance sites to globe: Synthesis and evaluation of the FLUXCOM approach. *Biogeosciences* 17, 1343–1365. <https://doi.org/10.5194/bg-17-1343-2020>
- Keeling, C.D., Whorf, T.P., Wahlen, M., van der Plichtt, J., 1995. Interannual extremes in the rate of rise of atmospheric carbon dioxide since 1980. *Nature*. <https://doi.org/10.1038/375666a0>
- Le Quéré, C., Andrew, R.M., Friedlingstein, P., Sitch, S., Pongratz, J., Manning, A.C., Ivar Korsbakken, J., Peters, G.P., Canadell, J.G., Jackson, R.B., Boden, T.A., Tans, P.P., Andrews, O.D., Arora, V.K., Bakker, D.C.E., Barbero, L., Becker, M., Betts, R.A., Bopp, L., Chevallier, F., Chini, L.P., Ciais, P., Cosca, C.E., Cross, J., Currie, K., Gasser, T., Harris, I., Hauck, J., Haverd, V., Houghton, R.A., Hunt, C.W., Hurtt, G., Ilyina, T., Jain, A.K., Kato, E., Kautz, M., Keeling, R.F., Klein Goldewijk, K., Körtzinger, A., Landschützer, P., Lefèvre, N., Lenton, A., Lienert, S., Lima, I., Lombardozi, D., Metzl, N., Millero, F., Monteiro, P.M.S., Munro, D.R., Nabel, J.E.M.S., Nakaoka, S.I., Nojiri, Y., Antonio Padin, X., Pregon, A., Pfeil, B., Pierrot, D., Poulter, B., Rehder, G., Reimer, J., Rödenbeck, C., Schwinger, J., Séférian, R., Skjelvan, I., Stocker, B.D., Tian, H., Tilbrook, B., Tubiello, F.N., Laan-Luijkx, I.T.V., Werf, G.R.V., Van Heuven, S., Viovy, N., Vuichard, N., Walker, A.P., Watson, A.J., Wiltshire, A.J., Zaehle, S., Zhu, D., 2018. Global Carbon Budget 2017. *Earth Syst. Sci. Data* 10, 405–448. <https://doi.org/10.5194/essd-10-405-2018>
- Lyon, B., 2004. The strength of El Niño and the spatial extent of tropical drought. *Geophys. Res. Lett.* 31, 1–4. <https://doi.org/10.1029/2004GL020901>
- Peylin, P., Law, R.M., Gurney, K.R., Chevallier, F., Jacobson, A.R., Maki, T., Niwa, Y., Patra, P.K., Peters, W., Rayner, P.J., Rödenbeck, C., Van Der Laan-Luijkx, I.T., Zhang, X., 2013. Global atmospheric carbon budget: Results from an ensemble of atmospheric CO₂ inversions. *Biogeosciences* 10, 6699–6720. <https://doi.org/10.5194/bg-10-6699-2013>
- Piao, S., Wang, X., Wang, K., Li, X., Bastos, A., Canadell, J.G., Ciais, P., Friedlingstein, P., Sitch, S., 2020. Interannual variation of terrestrial carbon cycle: Issues and perspectives. *Glob. Chang. Biol.* 26, 300–318. <https://doi.org/10.1111/gcb.14884>
- Poulter, B., Frank, D., Ciais, P., Myneni, R.B., Andela, N., Bi, J., Broquet, G., Canadell, J.G., Chevallier, F., Liu, Y.Y., Running, S.W., Sitch, S., Van Der Werf, G.R., 2014. Contribution of semi-arid ecosystems to interannual variability of the global carbon cycle. *Nature* 509, 600–603. <https://doi.org/10.1038/nature13376>
- Rödenbeck, C., Zaehle, S., Keeling, R., Heimann, M., 2018. History of El Niño impacts on the global carbon cycle 1957–2017: A quantification from atmospheric CO₂ data. *Philos. Trans. R. Soc. B Biol. Sci.* 373. <https://doi.org/10.1098/rstb.2017.0303>
- Tan, Z.H., Zeng, J., Zhang, Y.J., Slot, M., Gamo, M., Hirano, T., Kosugi, Y., Da Rocha, H.R., Saleska, S.R., Goulden, M.L., Wofsy, S.C., Miller, S.D., Manzi, A.O., Nobre, A.D., De Camargo, P.B., Restrepo-Coupe, N., 2017. Optimum air temperature for tropical forest photosynthesis: Mechanisms involved and implications for climate warming. *Environ. Res. Lett.* 12. <https://doi.org/10.1088/1748-9326/aa6f97>
- Van Der Werf, G.R., Randerson, J.T., Giglio, L., Van Leeuwen, T.T., Chen, Y., Rogers, B.M., Mu,

- M., Van Marle, M.J.E., Morton, D.C., Collatz, G.J., Yokelson, R.J., Kasibhatla, P.S., 2017. Global fire emissions estimates during 1997-2016. *Earth Syst. Sci. Data* 9, 697–720.
<https://doi.org/10.5194/essd-9-697-2017>
- Wang, J., Zeng, N., Wang, M., 2016. Interannual variability of the atmospheric CO₂ growth rate: Roles of precipitation and temperature. *Biogeosciences* 13, 2339–2352.
<https://doi.org/10.5194/bg-13-2339-2016>
- Wang, W., Ciais, P., Nemani, R.R., Canadell, J.G., Piao, S., Sitch, S., White, M.A., Hashimoto, H., Milesi, C., Myneni, R.B., 2013. Variations in atmospheric CO₂ growth rates coupled with tropical temperature. *Proc. Natl. Acad. Sci.* 110, 13061–13066.
<https://doi.org/10.1073/pnas.1219683110>
- Wang, X., Piao, S., Ciais, P., Friedlingstein, P., Myneni, R.B., Cox, P., Heimann, M., Miller, J., Peng, S., Wang, T., Yang, H., Chen, A., 2014. A two-fold increase of carbon cycle sensitivity to tropical temperature variations. *Nature* 506, 212–215.
<https://doi.org/10.1038/nature12915>
- Yue, C., Ciais, P., Houghton, R.A., Nassikas, A.A., 2020. Contribution of land use to the interannual variability of the land carbon cycle. *Nat. Commun.* 11, 3170.
<https://doi.org/10.1038/s41467-020-16953-8>

REVIEWER COMMENTS

Reviewer #1 (Remarks to the Author):

The authors have made great efforts to address my comments, which are much appreciate. I still have a few concerns/suggestions which I hope the authors can address.

1. The revised title does not seem to be informative. "global carbon cycle variability" is quite vague, and "associated" can have different meanings. It'd be great if the authors can improve the accuracy, clarify, and information content of the title.

2. The causal relationship among STD_CGR, temperature sensitivity, and drought area are not really clear. What is the cause, what is the effect? The analyses in this paper is purely based on correlation, while the description is here and there. For example, in line 132-135, the first statement indicates that the STD_CGR is the cause while temperature sensitivity is the outcome; while the following sentence indicated that the temperature sensitivity is the cause. The logics seems to be self-contradicting.

3. I don't quite understand why NEE map can be used as the spatial weighting. There is a lack of sufficient justification.

4. This is the most concerning one. My impression is that after the removal of autocorrelation, the strong R2 tends to disappear, although p value is still somewhat significant, e.g., Fig. S3 and Fig.

1. However, the author still claimed that removal of autocorrelation does not affect the conclusion, which I personally don't think it is the case.

Reviewer #2 (Remarks to the Author):

The authors have fully addressed my reviews and I consider the work to be noteworthy and ready for publication.

Reviewer #3 (Remarks to the Author):

I have reviewed the response to reviewers and re-read the manuscript, and it seems to me the authors have been able to adequately address the concerns I raised in my previous review.

Tropical extreme droughts drive long-term increase in atmospheric CO₂ growth rate variability

NCOMMS-21-08853A Response to Reviewers

Authors: We are grateful to the support and constructive comments from the reviewers, and the invitation from the editor to revise our manuscript. We have carefully followed the reviewer and editor's comments to clarify our methods and conduct additional analyses, in particular to show the robustness of the causal relationship. Please see our point-to-point response below in blue text.

REVIEWER COMMENTS

Reviewer #1 (Remarks to the Author):

The authors have made great efforts to address my comments, which are much appreciate. I still have a few concerns/suggestions which I hope the authors can address.

Authors: We appreciate the continued interest from the reviewer in our manuscript and address their remaining concerns below.

R1C1: 1. The revised title does not seem to be informative. "global carbon cycle variability" is quite vague, and "associated" can have different meanings. It'd be great if the authors can improve the accuracy, clarify, and information content of the title.

Authors: Thank you, we agree and have revised the title to "Tropical extreme droughts drive long-term increase in atmospheric CO₂ growth rate variability" based on your suggestion. This change emphasizes atmospheric CO₂ growth rate variability rather than the global carbon cycle, which is more appropriate as growth rate variability is the focus of our study.

R1C2: 2. The causal relationship among STD_CGR, temperature sensitivity, and drought area are not really clear. What is the cause, what is the effect? The analyses in this paper is purely based on correlation, while the description is here and there. For example, in line 132-135, the first statement indicates that the STD_CGR is the cause while temperature sensitivity is the outcome; while the following sentence indicated that the temperature sensitivity is the cause. The logics seems to be self-contradicting.

Authors: In our study, we report that the increase in tropical drought area amplified the variability in atmospheric CO₂ growth rate (STD_{CGR}), which was further responsible for the previously observed increase in apparent climate sensitivities of CGR (Wang et al., 2014).

Note that our findings differ from those of other studies, which implied that the climate sensitivities of CGR drove the variation in CGR (Humphrey et al., 2018; Wang et al., 2013), because we found the so-called climate sensitivities of CGR are largely an apparent characteristic of the global carbon cycle. The climate sensitivities themselves are highly sensitive to the methods and datasets used for their derivation (Fig. S1), and are also apparent in that they can appear to change if drought area changes, as we show.

We were unable to locate the statement mentioned by the reviewer in L132-135. Perhaps the reviewer meant a statement on L138 regarding the model results for STD_{NEE}, which we have now removed to avoid potential confusion. We have carefully revised the manuscript to make sure our reasoning is consistent and clear throughout.

R1C3: 3. I don't quite understand why NEE map can be used as the spatial weighting. There is a lack of sufficient justification.

Authors: We present the justification for using the FLUXCOM NEE map as a spatial weight in our methods section (L448-454):

“Quantifying the regional contributions to STD_{CGR}. Based on the tight correlation between extreme drought-affected area and STD_{CGR}, we adopted a time-for-space substitution to quantify regional (i.e., tropical America, tropical Africa, tropical Asia, semi-arid and forests) contributions to STD_{CGR}, using the drought-affected area detected for these regions. To consider the spatial variation of regions in their land-atmospheric CO₂ exchange capacity, we used the multi-year average FLUXCOM NEE map as the spatial weight, as FLUXCOM NEE represents our best estimate of the spatial variation in NEE (Jung et al., 2020) and the product shows potential in capturing the extreme drought influence on NEE (Fig. 2d; Fig S5).”

Following the comment from the reviewer, we have added the statement below in the manuscript to explain why using FLUXCOM NEE is the preferred available solution to identify regional contributions to STD_{CGR} (L456-468).

“We note that to estimate the regional contribution of extreme droughts to STD_{CGR}, it is necessary to account for all transient and long-term carbon fluxes incurred by extreme droughts. In theory this can be simulated by process-based DGVMs. But current DGVMs, as our results suggest (Fig. 2d), inadequately capture the drought impacts on STD_{CGR}. Another option is to use remotely sensed aboveground biomass (AGB) (Baccini et al., 2017) as the spatial weight [as reviewer 1 suggested in the last round]. However, this approach would implicitly assume that extreme droughts induced carbon losses proportionally to AGB. This assumption is questionable as drylands have less biomass but usually comparable net carbon exchange to wet

forests (Ahlström et al., 2015; Poulter et al., 2014). Therefore, using the data-driven net flux product (i.e., FLUXCOM NEE) as a weight is the preferred available option to approximate the regional contribution to STD_{CGR} , as FLUXCOM NEE showed potential in capturing the extreme drought influence on NEE (Fig. 2d; Fig S5). Importantly, our study is designed to provide a first-order quantification of the regional contribution to STD_{CGR} at coarse continental and biome scales, while the fine-scale variations at the pixel-level within each continent and biome remain to be addressed.”

R1C4: 4. This is the most concerning one. My impression is that after the removal of autocorrelation, the strong R2 tends to disappear, although p value is still somewhat significant, e.g., Fig. S3 and Fig. 1. However, the author still claimed that removal of autocorrelation does not affect the conclusion, which I personally don't think it is the case.

Authors: The removal of autocorrelation using the Cochrane-Ocrutt procedure is expected to cause a lower r^2 , since the procedure largely eradicates short-term variability in CGR. It is the derived sensitivity however that is important. For example, a time series $T = V_s + V_l + \text{noise}$, where V_s refers to short-term variability and V_l refers to long-term variability. With the removal of V_s in time series, we are likely to see a decrease in the variability of T and thus the signal/noise ratio for T , which causes the decrease in r^2 in the correlation analysis. We are confident in our conclusion however, as even after the removal of autocorrelation, our results still hold a high level of significance (Fig. S3) and importantly, the sensitivity of STD_{CGR} to drought-affected area is consistent before and after the Cochrane-Ocrutt correction (Fig. S3b vs. Fig. 2c). We now however add an additional independent autocorrelation analysis (see below), which further supports the robustness of our results.

We acknowledge the changes in r^2 after the correction in the results as follows (L146-148) – “Note that the removal of autocorrelation resulted in a smaller deterministic coefficient (r^2) between long-term water availability and STD_{CGR} , which is expected given the correction removes all the short-term variation in time series.”

We also need to clarify that the Cochrane-Ocrutt procedure we implemented is a very strong autocorrelation correction, which renders a conservative estimate of the drought area - STD_{CGR} correlation (Fig. S3 a, b). In comparison, we implemented **an alternative autocorrelation correction** that does not rely on reducing short-term variability in the time series. In this new approach, we divided the 58-year records into twelve independent 5-year segments (only the last segment has 2-year overlap with its previous segment), and calculated γ_{CGR}^T , γ_{CGR}^W and STD_{CGR} and the drought-affected area for each 5-year segment. Each 5-year segment is therefore independent from others since there is no overlap between them.

Using this approach, we found that the correlations between drought area and STD_{CGR} , and the correlation between γ_{CGR}^T and STD_{CGR} are still highly significant (Fig. S3c,d). They support our reasoning of extreme drought -> STD_{CGR} -> the apparent temperature sensitivity of CGR. We have added the statements above in L189-191 and L495-500. The r^2 and derived slope differ

from what we obtained after the Cochrane-Ocrutt procedure primarily because this new test has much fewer samples for the correlation analysis, and the new test was not as strict as the Cochrane-Ocrutt procedure in terms of removing all short-term variation (i.e., coefficients from each 5-year segment might be still weakly related though they are independent samples). The two corrections we implemented both support our conclusion that extreme droughts were responsible for the changes in long-term variability in atmospheric CO₂ growth rate.

PS: in our re-examination of the materials, we found an error in the data plotted in Fig. S3a. We have corrected it in the current version. The correction does not affect our conclusion, but improves issues with the graphical representation of the results which may have contributed to the reviewer's sense that the autocorrelation removal degraded the results significantly.

Fig. S3 | Remove the autocorrelations in Fig. 1d and Fig. 2c. We used two methods to evaluate the impact of autocorrelation. In (a) and (b), we used the Cochrane-Ocrutt procedure (see Methods) to remove autocorrelations in time series and obtain adjusted climate sensitivities of CGR (i.e., $\text{adj. } \gamma_{\text{CGR}}^{\text{T}}$ and $\text{adj. } \gamma_{\text{CGR}}^{\text{W}}$) and adjusted STD_{CGR} . In (c) and (d), we divided the time series of CGR into 12 independent 5-year segments and derive $\gamma_{\text{CGR}}^{\text{T}}$, $\gamma_{\text{CGR}}^{\text{W}}$ and STD_{CGR} for each 5-year segment to obtain non-autocorrelated time series of climate sensitivities and STD_{CGR} . (a,c) The relationships between climate sensitivities of CGR and STD_{CGR} (shading: 95% confidence

interval); (b) The relationship between STD_{CGR} and extreme drought-affected area (shading: 95% confidence interval).

Reviewer #2 (Remarks to the Author):

The authors have fully addressed my reviews and I consider the work to be noteworthy and ready for publication.

Authors: We appreciate the support from the reviewer and are happy to hear that they feel the paper is now suitable for publication.

Reviewer #3 (Remarks to the Author):

I have reviewed the response to reviewers and re-read the manuscript, and it seems to me the authors have been able to adequately address the concerns I raised in my previous review.

Authors: We thank the reviewer for helping us improve the manuscript and for recommending publication.

REVIEWERS' COMMENTS

Reviewer #1 (Remarks to the Author):

The authors have attempted to address my comments. The manuscript can now be accepted.